# Agonistic CD27 antibody potency is determined by epitope-dependent receptor clustering augmented through Fc-engineering

Franziska Heckel [1,2], Anna H. Turaj[1,2], Hayden Fisher[1,2,3,4], H. T. Claude Chan[1,2], Michael J. E. Marshall [1,2], Osman Dadas[1,2], Christine A. Penfold[1,2], Tatyana Inzhelevskaya[1,2], C. Ian Mockridge[1,2], Diego Alvarado[5], Ivo Tews[3,4], Tibor Keler[5], Stephen A. Beers [1,2], Mark S. Cragg [1,2] & Sean H. Lim [1,2 ✉]

Agonistic CD27 monoclonal antibodies (mAb) have demonstrated impressive anti-tumour efficacy in multiple preclinical models but modest clinical responses. This might reflect current reagents delivering suboptimal CD27 agonism. Here, using a novel panel of CD27 mAb including a clinical candidate, we investigate the determinants of CD27 mAb agonism. Epitope mapping and *in silico* docking analysis show that mAb binding to membrane-distal and external-facing residues are stronger agonists. However, poor epitope-dependent agonism could partially be overcome by Fc-engineering, using mAb isotypes that promote receptor clustering, such as human immunoglobulin G1 (hIgG1, h1) with enhanced affinity to Fc gamma receptor (FcγR) IIb, or hIgG2 (h2). This study provides the critical knowledge required for the development of agonistic CD27 mAb that are potentially more clinically efficacious.

[1] Centre for Cancer Immunology, Faculty of Medicine, University of Southampton, Southampton SO16 6YD, UK. [2] Cancer Research UK Research Centre, Faculty of Medicine, University of Southampton, Southampton SO16 6YD, UK. [3] Institute for Life Sciences, University of Southampton, Highfield Campus, Southampton SO17 1BJ, UK. [4] Biological Sciences, University of Southampton, Highfield Campus, Southampton SO17 1BJ, UK. [5] Celldex Therapeutics, Inc., Hampton, NJ 08827, USA. ✉email: s.h.lim@soton.ac.uk

mmunotherapy is becoming central to the development of new and more effective treatments for cancer. In particular, the advent of direct tumour targeting mAb like rituximab and trastuzumab, and immunomodulatory mAb which block immune checkpoints such as PD-1, PD-L1 or CTLA-4, have demonstrated impressive clinical efficacy[1–5]. Nevertheless, further improvement is required as a significant proportion of cases continue to be resistant. Agonistic mAb targeting co-stimulatory molecules of the tumour necrosis factor receptor superfamily (TNFRSF) (e.g. OX40, CD40, 4-1BB, GITR or CD27) are attractive candidates for cancer immunotherapy. However, unlike checkpoint inhibitors where the role of the mAb is simply to block the ligand from binding the receptor, the requirements for agonistic receptor stimulation are more complex and likely differ depending on the target.

To date, multiple agonistic mAb (e.g. directed to CD40, OX40, 4-1BB, GITR, CD27, CD28 and ICOS) have entered clinical trials but objective responses have been modest[6–8]. We reason that this is due to an ongoing lack of understanding of the mechanisms that govern agonism and how best to leverage activity without toxicity in humans leading to the development and clinical translation of suboptimal mAb formats. Tumour necrosis factor receptors (TNFR) are characterised by their extracellular cysteine-rich domains (CRD). The TNFR ligands such as OX40L[9], 4-1BBL[10,11], GITRL[12] and CD40L[13] form homotrimers via a C-terminal TNF-homology domain. Assembly of the trimeric ligand with its respective TNFR results in the recruitment of TNFR-associated factors (TRAF) to TRAF-interaction motifs contained within the intracellular domain of each TNFR and subsequent triggering of downstream signalling pathways such as the NF-κB or MAP kinase pathway[14]. Agonistic mAb directed to many TNFR can mimic, and in some instances, enhance ligand activity to promote powerful biological responses[15–18]. Current evidence indicates that for CD40, OX40 and 4-1BB, mAb isotype and epitope specificity are critical for governing activity but that the requirements are unique for each receptor. In mice, CD40, OX40 and 4-1BB multimerisation and consequently agonism, can be achieved through Fc:FcγR interaction by employing a murine IgG1 (m1) isotype which has affinity for the murine inhibitory FcγR, FcγRIIb, and activatory FcγRIII[19]. Similarly, mutation of the Fc domain of h1 to enhance engagement for the human inhibitory FcγR, FcγRIIb (SE/LF, V9 or V11 mutations)[20–23] also confers increased agonism for TNFR. FcγR-independent receptor cross-linking can also be achieved by employing a hIgG2B (h2B) isotype. Here, it is hypothesised that the specific disulphide bond arrangement in the hinge links directly to the F(ab) arms to produce a more rigid conformation. For CD40 mAb, this results in increased receptor clustering and enhanced agonism through activation of antigen presenting cells and consequently CD8$^+$ T-cell expansion[18,24–27].

Although it is clear that mAb isotype plays a central role in providing agonism, it is also evident that epitope location matters. Murine OX40 mAb binding to membrane-proximal epitopes mediate superior agonism over those binding more distally[28], with similar findings for human OX40 (hOX40) mAb[29]. In contrast, for hCD40 mAb, strong agonism is observed with mAb binding to the most distal epitopes[18]. Even so, it is clear fine specificity dictates the extent of agonism, with CP870,893 evoking more powerful responses than ChiLob 7/4, despite both binding within CRD1 of CD40[18].

We previously demonstrated that stimulation of the TNFRSF member, CD27 on T cells, promotes activation of bystander myeloid cells leading to increased antibody-dependent cellular phagocytosis (ADCP), in the context of a second tumour-targeting mAb[16]. We hypothesised that the efficacy of this combination is contingent upon the use of a sufficiently robust CD27 agonist. Varlilumab is a fully human agonistic h1 mAb that is furthest in clinical development amongst CD27 mAb[30,31]. In vivo data using murine Fc bearing versions of varlilumab indicate that m1 demonstrated better anti-tumour efficacy than mIgG2a (m2a) in a lymphoma model, whereas the reverse was true for subcutaneously implanted solid tumours[32]. Direct examination of the tumours was not undertaken to explain the potentially differing modes of action between m1 and m2a, but nevertheless it provided the rationale for selecting a h1 isotype for clinical development. In addition, it provided further indication that mAb isotype is a key determinant of CD27 mAb therapy but that this requirement may be context-dependent.

However, it remains to be explored as to how the optimal therapeutic efficacy of anti-CD27 can be achieved. Here, we investigated the influence of mAb isotype and epitope specificity in agonist CD27 mAb therapy for cancer. We found that the efficacy of depleting mAb such as anti-CTLA-4 and anti-CD25 can be enhanced by CD27 mAb in a murine colon adenocarcinoma tumour model. Using a panel of novel CD27 mAb, we demonstrated that mAb binding to membrane-distal, externally-facing epitopes were more agonistic than mAb binding to membrane-proximal, internally-facing epitopes. Further, weakly agonistic mAb could be improved by Fc-engineering, specifically by employing mAb of h2 isotype or with mutations that enhance Fc binding to the inhibitory FcγRIIb.

## Results

**CD27 stimulation enhances Treg depletion in CT26.** Before exploring how best to optimise CD27 agonism and revealing the key mAb determinants, we wished to expand the general therapeutic areas where this might be of use. Using an aggressive murine B-cell lymphoma model, BCL$_1$, we previously demonstrated that anti-CD20 combined with anti-CD27 produced highly effective tumour control[16]. Based on these findings, we hypothesised that agonistic anti-CD27 could also enhance other, non-CD20, depleting mAb in solid tumour models. To investigate this, we combined anti-mouse CD27 (mCD27) with anti-mCTLA-4[33,34] in the CT26 colorectal cancer model (Fig. 1a). Administration of anti-mCTLA-4 or anti-mCD27 delayed tumour growth compared to the isotype control, but did not elicit long-term survival whereas the mAb combination cured 40% (4/10) of mice (Fig. 1b). The median survival in the combination (39 days) was significantly greater than the control (22.5 days), and both monotherapy arms, anti-mCTLA-4 (24 days) and anti-mCD27 (27days) (Fig. 1c).

Anti-CTLA-4 mediates its anti-tumour activity in part, by depleting tumour-infiltrating Tregs[33,35]. In the CT26 model, we observed reduction of Tregs by anti-mCTLA-4 compared to the control arm (34.5% vs 45.2% of CD4$^+$ T cells, respectively) (Fig. 1d and Supplementary Fig. 1a). No reduction in percentage of Tregs was observed with anti-mCD27 (42.3% of CD4$^+$ T cells), but the combination of anti-mCD27 and anti-mCTLA-4 significantly enhanced Treg depletion (26.3% of CD4$^+$ T cells). Increased CD8$^+$ T-cell expansion was observed in the anti-mCD27 (74.4%) and combination (66.4%) arms (Fig. 1e and Supplementary Fig. 1a), leading to a markedly improved CD8/Treg ratio (>2-fold) in the combination versus either monotherapy (Fig. 1f).

We had previously observed anti-CD27-mediated activation of myeloid cells in the BCL$_1$ model[16]. In the CT26 model we observed a shift in the phenotype of the tumour-infiltrating myeloid cells upon anti-CD27 treatment, with CD11b$^+$ cells gaining F4/80 expression, indicative of macrophage differentiation[36] (Fig. 1g, h and Supplementary Fig. 1b). These cells also expressed 2-fold higher levels of the activatory FcγRIV

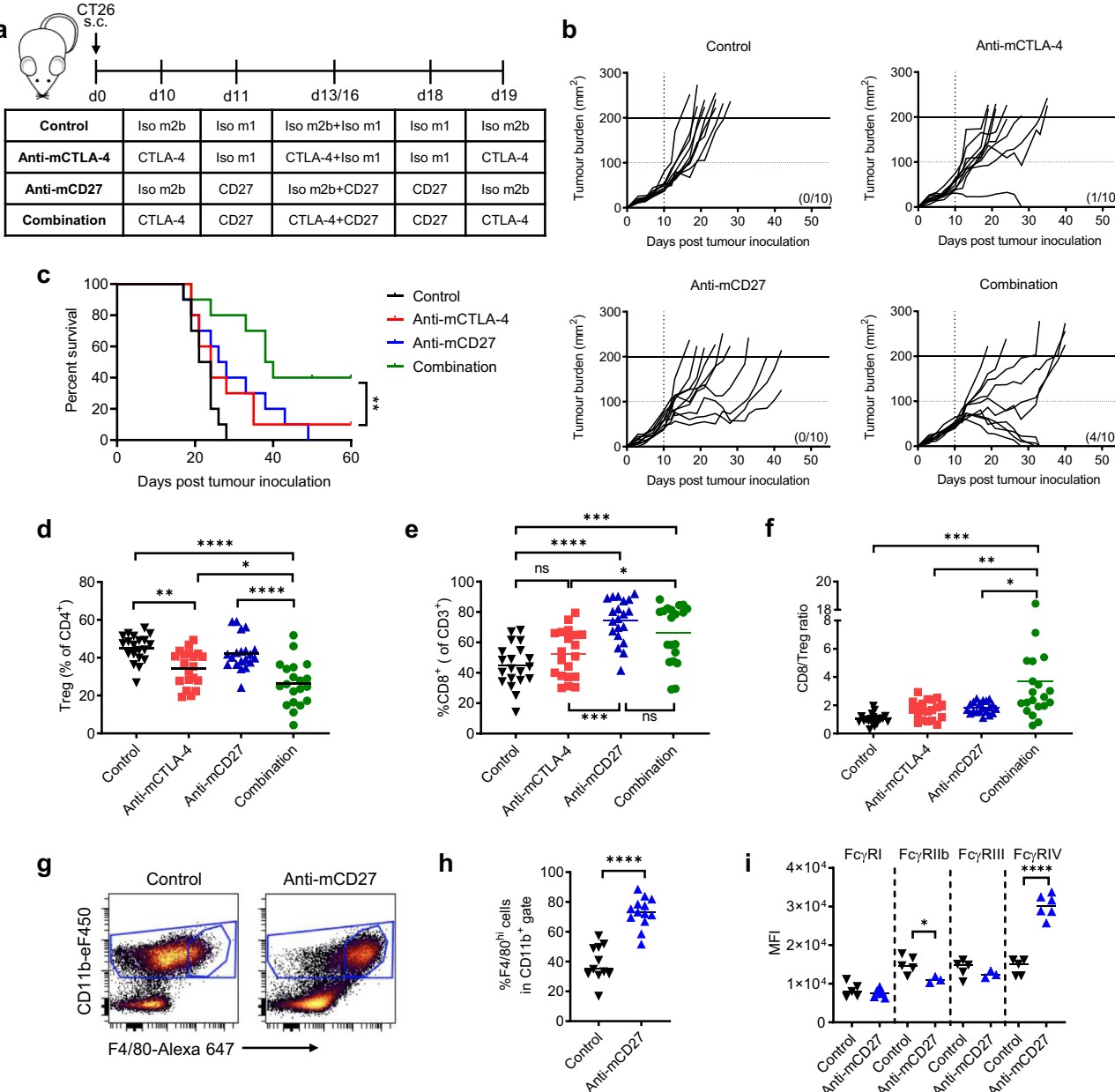

**Fig. 1 CD27 stimulation enhances mAb-mediated Treg depletion leading to improved survival. a** CT26-bearing BALB/c mice treated with either anti-mCTLA-4 m2b (9D9, 200 μg) on days 10, 13, 16 and 19 or anti-mCD27 m1 (AT124-1, 100 μg) on days 11, 13, 16 and 18, or in combination. **b** Tumour growth in individual mice treated as described in **a** ($n = 10$ per group from two independent experiments). **c** Survival of mice treated in **a** from two independent experiments ($n = 10$ per group). Log-rank test was used to assess p-values; **$p < 0.01$. **d**–**f** CT26-bearing mice were treated as described in **a**. Tumours were harvested on day 20 and the **d** %Tregs and **e** %CD8+ T cells and **f** CD8/Treg ratio was determined. Graphs show data from four independent experiments ($n = 20$ per group). Data were assessed using one-way ANOVA and Tukey's test; *$p < 0.05$, **$p < 0.01$, ***$p < 0.001$, ****$p < 0.0001$. See also Supplementary Fig. 1. **g** Representative flow cytometry dot plots of F4/80 expression on intra-tumoural CD11b+ cells within the CT26 tumours described in **d**. **h** Percentage of CD11b+F4/80hi intra-tumoural immune cells of CD11b+ cells as described in Supplementary Fig. 1. **i** Expression of FcγRI, FcγRIIb, FcγRIII and FcγRIV on CD11b+F4/80hi cells of CD11b+ cells. Two-tailed Student's t-test was used to assess p-values; *$p < 0.05$, ****$p < 0.0001$.

and displayed a reduction of FcγRIIb. Minimal changes were observed in the expression of the other FcγRs (Fig. 1i).

When we substituted anti-mCTLA-4 with another Treg-depleting mAb, anti-mCD25 (PC61), Treg depletion was also significantly enhanced by addition of anti-mCD27 (Supplementary Fig. 1c). Altogether, these results support the general applicability of using agonistic anti-CD27 to enhance the effects of Treg-depleting mAb in different tumour models. Having expanded on the potential broad applicability of this therapeutic approach we explored how best to optimise the CD27 mAb-mediated agonism.

**The therapeutic efficacy of agonistic anti-CD27 is dependent on mAb isotype.** To examine the contribution of mAb isotype to anti-CD27 therapy, we returned to our well-validated lymphoma model and compared m1 or m2a variants of anti-mCD27. m1 binds to FcγRIIb and FcγRIII without appreciable affinity for other FcγR, similar to the rat IgG2a (r2a) mAb we previously employed, whilst m2a engages far more strongly with activatory FcγRs and preferentially binds to FcγRI and FcγRIV (Supplementary Fig. 2). Anti-mCD27 m1 or m2a were combined with anti-CD20 in BCL1-bearing mice (Fig. 2a, b). Here, the isotype-

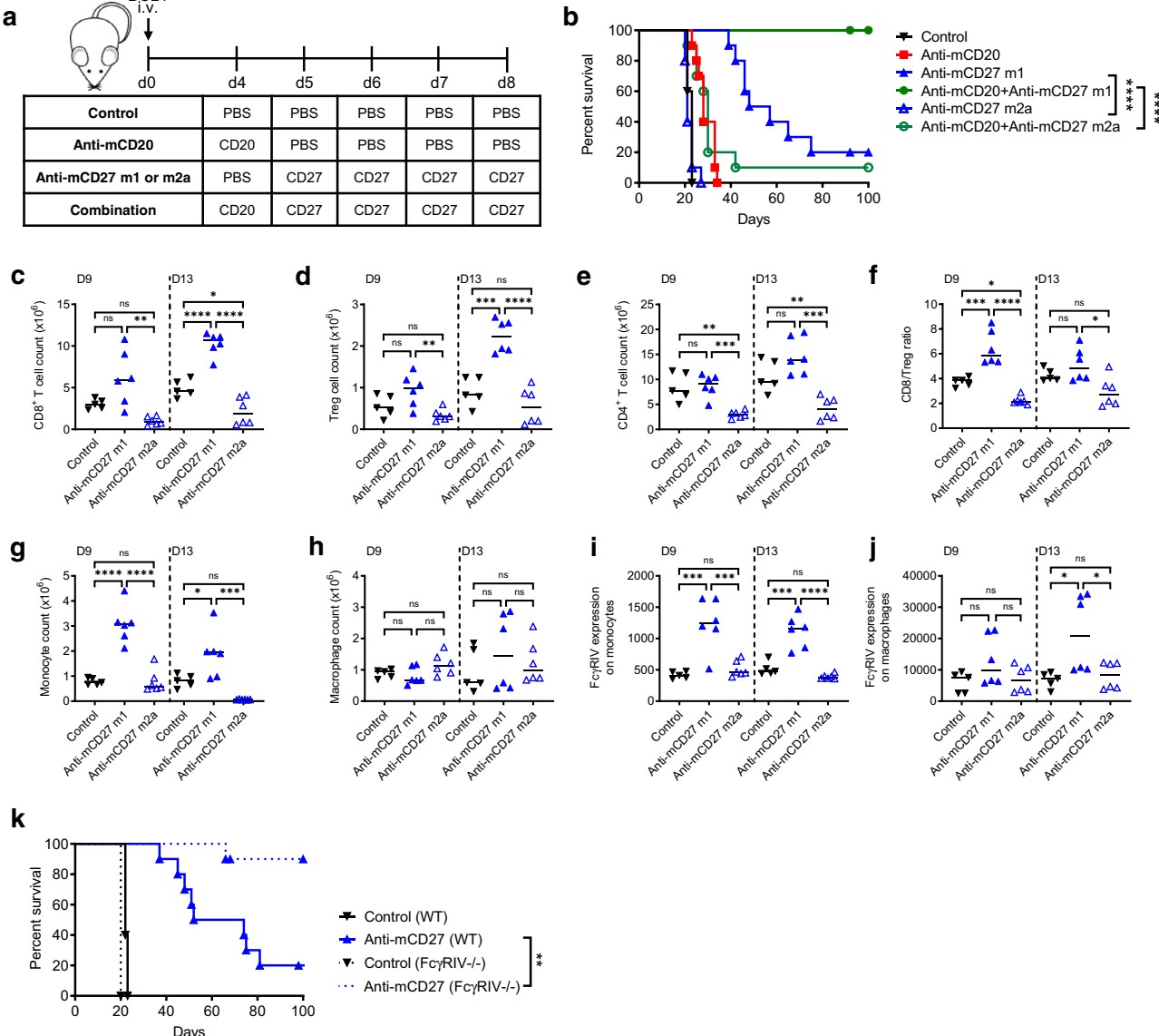

**Fig. 2 The efficacy of anti-CD27 is strictly isotype dependent in the BCL₁ mouse model. a** BCL₁-bearing BALB/c mice were treated with either PBS (control), anti-mCD20 m2a (18B12, 200 μg) on day 4, anti-mCD27 m1 (AT124-1, 50 μg) or anti-mCD27 m2a (AT124-1, 50 μg) on days 5–8 or in combination. **b** Survival of mice treated as described in **a**. Graph shows $n = 10$ per group, compiled from two independent experiments. Data were assessed using Log-rank test; ****$p < 0.0001$. **c–j** BCL₁-bearing BALB/c mice were treated with anti-mCD27 m1 (AT124-1, 50 μg) or anti-mCD27 m2a (AT124-1, 50 μg) or with PBS (control) on days 5–8. Tumours were harvested on days 9 and 13. Cell counts were determined, and cells analysed using flow cytometry. Graphs show total cell counts of **c** CD8+ T cells, **d** Tregs, **e** CD4+ T cells and **f** CD8/Treg ratio, total numbers of **g** monocytes and **h** macrophages and the FcγRIV expression on **i** monocytes and **j** macrophages. Graphs are representative of one experiment with $n = 5$–6 per group. Shown are medians and one-way ANOVA and Tukey's test were used to assess $p$-values; *$p < 0.05$, **$p < 0.01$, ***$p < 0.001$, ****$p < 0.0001$. See also Supplementary Fig. 3. **k** Survival of BCL₁-bearing WT or FcγRIV$^{-/-}$ mice treated with PBS (control) or anti-mCD27 r2a (AT124-1, 50 μg) on days 5–8. Graph shows $n = 10$ per group, representative of four independent experiments. Log rank test was used to assess $p$-values; **$p < 0.01$.

treated mice had a median survival of 23 days and the anti-CD20 treated mice a median survival of 28 days. Anti-mCD27 m1 monotherapy, and combination with anti-CD20 produced median survivals of 52.5 and beyond 100 days, respectively, similar to previous data using r2a[16]. In contrast, anti-mCD27 m2a monotherapy (median survival 21 days), or combination with anti-CD20 (median survival 28 days), had a negligible effect on prolonging survival.

To understand the mechanism behind the differences between m1 and m2a anti-mCD27, tumours were harvested on days 9 and 13 post BCL₁-inoculation. CD8+ T cells were significantly elevated after treatment with anti-mCD27 m1 (d9: 2-fold, d13:

2.3-fold) but reduced after anti-CD27 m2a therapy (d9: 3.4-fold, d13: 2.5-fold) (Fig. 2c and Supplementary Fig. 3). In parallel, m1 treatment also led to increased Tregs (d13: 2.7-fold) and conversely, anti-mCD27 m2a treatment induced a trend towards reduced Tregs (Fig. 2d). In the CD4+ T-cell compartment, significant depletion by m2a was observed (day 9: 2.6-fold, day 13: 2.3-fold) but no significant changes were seen with anti-mCD27 m1 (Fig. 2e). Overall, treatment of mice with anti-mCD27 m1, but not m2a, significantly improved the CD8/Treg ratio, and this was most prominent at day 9 (Fig. 2f).

In the myeloid compartment, anti-mCD27 m1 produced changes comparable to those previously observed with r2a

(Fig. 2g–j). A 3-fold increase was observed in infiltrating monocytes on days 9 and 13 (Fig. 2g) with minimal change in macrophage numbers after m1 treatment (Fig. 2h). In contrast, treatment with m2a led to a reduction of monocytes on d13. Anti-mCD27 m1 also resulted in upregulation of FcγRIV on both monocytes (Fig. 2i) and macrophages (Fig. 2j). In comparison, minimal change was observed in macrophage numbers, or FcγRIV, with anti-mCD27 m2a.

Thus, in this model, the anti-tumour activity of anti-mCD27 is dependent on appropriate FcγR interaction, with engagement of activatory FcγR detrimental to therapy, likely as a result of depletion of CD8[+] and effector CD4[+] T cells. To validate this hypothesis, mice lacking FcγRIV, recognised to play a central role in cell depletion via ADCP[37–40], were inoculated with BCL$_1$ and treated with either an isotype control or anti-mCD27 (Fig. 2k). Here, the median survival of FcγRIV$^{-/-}$ mice was not reached at 100 days after treatment with the mCD27 mAb, compared to a median survival of 63 days with wild type (WT) mice, demonstrating an improvement in therapy in the absence of this FcγR.

**A new panel of hCD27 mAb that span a range of binding affinities**. The above pattern of response closely mirrors that seen with anti-CD40, where m1 but not m2a delivers powerful agonism and anti-tumour effects[41,42]. With anti-CD40 mAb both epitope and isotype contribute to agonism[18]. To explore these dual aspects with regards to human CD27 (hCD27), we generated a new panel of mAb targeting hCD27 (AT133-2, AT133-5, AT133-11 and AT133-14) and compared them to another commercially patented agonist, hCD27.15[43] and the clinically relevant mAb varlilumab (varli)[44,45] (Fig. 3).

Differences in maximum specific binding (Bmax) on CD4[+] T cells were observed by flow cytometry (Fig. 3a, b). The lowest Bmax was detected with hCD27.15, which was 5-fold lower than varli, AT133-2, AT133-5 and AT133-14, which all had comparable Bmax. All mAb reached saturation at 1 μg/ml, except AT133-11, which saturated at 4 μg/ml. To assess bivalent mAb binding avidity, surface plasmon resonance (SPR) was employed (Fig. 3c, d). Here varli, AT133-2 and AT133-14 demonstrated the highest affinity ($K_D$ 3.0, 5.7, 0.6 × 10$^{-9}$, respectively), followed by AT133-5 (28.2 × 10$^{-9}$), and hCD27.15 (30.8 × 10$^{-9}$), then AT133-11 (50.4 × 10$^{-9}$). All the mAb had relatively high association rates, but the dissociation rates were variable with hCD27.15 having the highest dissociation rate (2.3 to 14.9-fold faster than other mAb).

**hCD27 mAb bind to a range of different epitopes within hCD27**. Next, we set out to identify the epitopes bound by these mAb (Fig. 4). To do so, we generated a range of truncation mutants of hCD27 expressing or lacking various CRD (Fig. 4a, b). These mutants were expressed transiently in 293 F cells, and binding of the mAb was detected with R-phycoerythrin-conjugated anti-hFc. All mAb and CD70, the ligand for CD27, bound best to WT hCD27 containing all three CRD (Fig. 4b). Binding to CRD1-containing hCD27 mutants was observed for AT133-2, AT133-5 and AT133-11, with AT133-14 restricted to CRD3-containing mutants. Varli only bound when CRD2 and CRD3 were present, and hCD27.15 and CD70 only bound to the WT molecule. The data suggests that AT133-2, AT133-5 and AT133-11 bind to CRD1, AT133-14 to CRD3 and varli within CRD2 and CRD3. The domain bound by hCD27.15 was not elucidated at this stage but its loss of binding to mutants may reflect its higher off-rate, leading to greater sensitivity to small structural perturbations.

To corroborate the above observations, a cross-blocking competition assay was performed using the mAb (Fig. 4c). Target cells were first incubated with anti-hCD27 m1 or CD70 m1 (initial reagent), and then subsequently with anti-hCD27 h1 or CD70 h1 (detected competing reagent). Binding of the competing mAb was examined with R-phycoerythrin-conjugated anti-hFc by flow cytometry. As expected all the CRD1-binding mAb (AT133-2, AT133-5 and AT133-11) cross-blocked each other, represented as no binding (dark blue) in the heatmap. AT133-14-binding was not blocked by any of the reagents tested, supporting its unique binding site within CRD3. Varli's binding was reduced by binding of AT133-14 and CD70, suggesting some overlap of respective epitopes. CD70's binding was poorly defined and either partially or fully blocked by a number of mAb (CRD1-binding AT133-5 and AT133-11, CRD2 and CRD3-binding varli and CRD3-binding AT133-14). Together with the truncation mutant data, this would suggest that CD70 has a broader contact surface with hCD27 than the mAb and that AT133-2 and hCD27.15 are both non-ligand blocking mAb that bind to distinct epitopes distal to CD70 (Supplementary Fig. 4a). For hCD27.15, only CRD1-binding AT133-2 blocked its binding. In addition, its binding was paradoxically increased by pre-incubation with AT133-11 (CRD1-binding) or CD70. This suggests that AT133-2 and hCD27.15 bind to membrane-distal epitopes and that binding of AT133-11 and CD70 might alter the conformation and arrangement of hCD27 on the cell surface, subsequently leading to increased binding of hCD27.15.

To further define the mAb binding sites, surface alanine scanning mutagenesis was undertaken by pairwise mutation of consecutive amino acids of hCD27 (Fig. 4d, f) and for further refinement, mutation of single amino acids within the pairwise mutations (Fig. 4g). The constructs were transiently expressed in 293 F cells and mAb binding was detected by flow cytometry using anti-hFc as before (Fig. 4e and Supplementary Fig. 4b, c). Residue pairs such as H43/Y44, M54/E56, T59/F60, L61/V62, G80/V81, S82/F83, N107/T109 and N113/E115 resulted in reduction of binding for all mAb except for AT133-14. The requirement for these residues for binding despite being spread across all three domains suggests that they might be critical for receptor conformation (Fig. 4f). Additional residues within hCD27 essential to the binding of each hCD27 mAb, were also identified. AT133-2, AT133-5 and AT133-11 required residues located in CRD1 (AT133-2: H68, R69; AT133-5: W45, K49; AT133-11: W45, Q47, G48, K49). The epitope for hCD27.15 was inferred to be located in CRD1 and CRD2 (reliant on K63 and D75), whereas varli appeared to require residues in CRD2 (R97) and AT133-14 in CRD3 (D126, K127). The residues indicated as crucial for binding of CD70 were located in CRD2 (R90, H92, E94). Taken together, the data indicate that these six hCD27 mAb target all three CRD of hCD27: CRD1: AT133-2, AT133-5, AT133-11, hCD27.15; CRD2: varli and CRD3: AT133-14. However, to gain more insight into their relative binding epitopes and in the absence of crystal structures we performed *in silico* docking analysis.

Docking of hCD27 with the mAb of interest allowed the generation of six proposed models of F(ab) binding (Fig. 5a–f and Supplementary Table 1). The docking was driven by information from the site-directed mutagenesis analysis, imposing restraints that guide the F(ab) CDR loops towards residues known to result in a loss of binding when mutated. The models for AT133-2, AT133-5 and AT133-11 all suggest similar binding orientations, contacting the membrane-distal portion of CRD1 resulting in the formation of an elongated mAb:antigen complex (Fig. 5a–c). The model for hCD27.15 (Fig. 5d) shows perpendicular binding with the majority of F(ab) heavy chain contacts occurring within CRD1, while the light chain also interacts with CRD2, in agreement with Fig. 4b. The model for binding of varli to hCD27 (Fig. 5e) shows the interaction occurring largely with CRD2, with

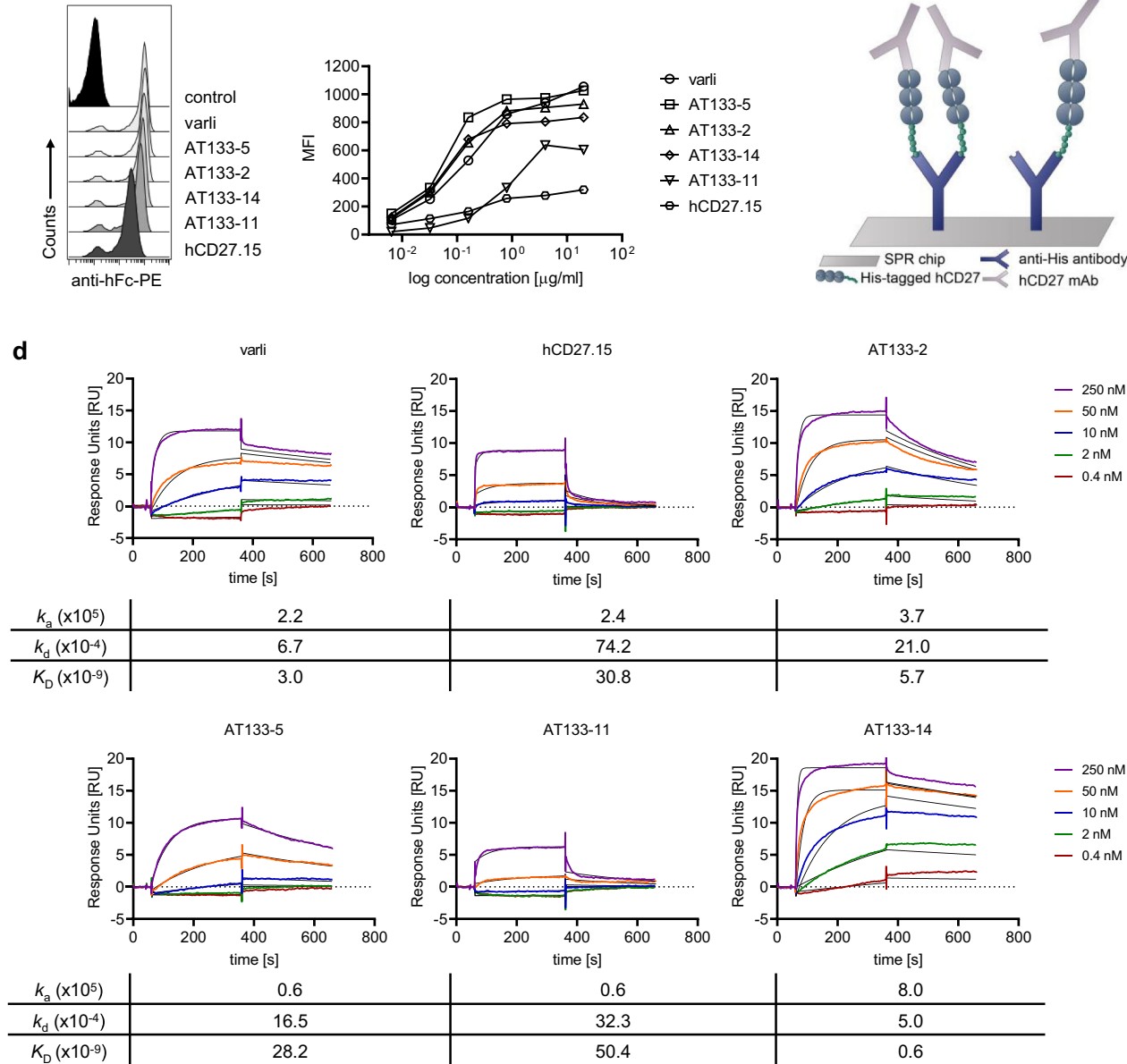

**Fig. 3 Characterisation of the binding of various hCD27 mAb. a** hCD27 mAb were added at specified concentrations to human peripheral blood mononuclear cells (PBMC) and binding to human CD4+ T cells detected by flow cytometry using R-phycoerythrin-conjugated anti-human Fc (anti-hFc). Representative histogram plots of anti-hCD27-binding to human CD4+ T cells at 4 μg/ml are shown. **b** Binding curves of the hCD27 h1 mAb to CD4+ T cells (as described in **a**). **c** Principle of SPR: Anti-His antibody was immobilised onto the SPR chip and His-tagged hCD27 captured. Following hCD27 mAb were injected and the interaction analysed using a Biacore system. **d** SPR analysis of hCD27 mAb: His-tagged hCD27 ligand was immobilised onto the SPR chip at 20 response units (RU). To allow saturation of hCD27 binding sites, the indicated mAb were flowed over the immobilised hCD27 for 300 s. A 1:1 binding fitting was applied.

a small number of contacts with CRD3, supporting the observations seen in the domain truncation analysis (Fig. 4b). For hCD27-AT133-14 (Fig. 5f), the greatest degree of interaction was observed with CRD3 compared to other models. The light chain forms a large interface with CRD3 while the heavy chain interfaces with CRD2, also supporting the observations in Fig. 4b. The crystal structure of the CD27-CD70 trimeric complex has recently been determined (PDB:7KX0[46]; Fig. 5g). The structure shows trimeric CD70 bound by three CD27 molecules, with CD70 interfacing CRD2 and CRD3 of CD27. This structure supports our site directed mutagenesis analysis, with critical contacts for complex formation being identified in CRD2 of CD27. Based on CD70's epitope with CD27, internal binding was

defined as CD70-facing, while external epitopes are on the opposite site of the receptor to the CD70 interface. Further, superimposition of the binding sites of CRD1-binding mAb (AT133-2, AT133-5, AT133-11 and hCD27.15) and CD70 demonstrate that AT133-5, AT133-11 and CD70 bind on the internal residues of hCD27 (Fig. 5h). Thus, while the epitopes bound by AT133-5 and AT133-11 are distinct to CD70, the mAb may sterically hinder the binding of CD70. In contrast, AT133-2 and hCD27.15 bind to external-facing residues within CRD1.

**Agonistic activity of hCD27 mAb is epitope-dependent but can be augmented by isotype selection.** Next, we characterised the

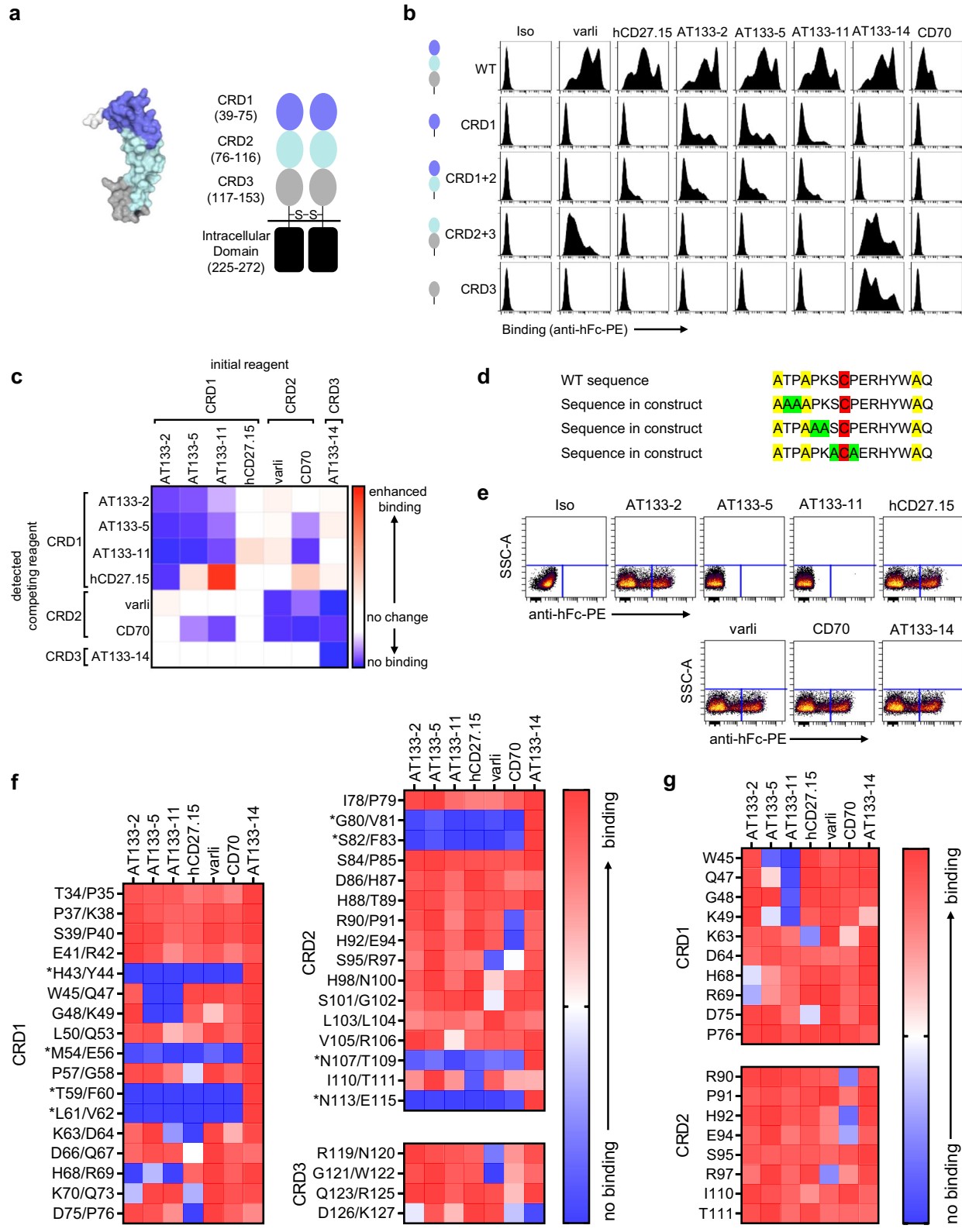

agonistic activity of these mAb (as h1 isotype) using a Jurkat NF-κB-GFP reporter cell line transfected with hCD27 (NF-κB-GFP hCD27 Jurkat) (Fig. 6a–d and Supplementary Fig. 5a). All mAb were agonistic, but evoked variable levels of GFP expression. hCD27.15 induced the highest level of GFP expression at 6 h (28.4% ± 9.6%) and 24 h (35.5% ± 17.4%) (Fig. 6b). AT133-2 was the second strongest agonist with %GFP of 23.8% ± 8.0% and

26.7% ± 13.8%, at 6 h and 24 h, respectively. Varli, AT133-5 and AT133-14 displayed comparable levels of stimulation at 6 h (varli: 20.2% ± 8.8%, AT133-5: 14.1% ± 5.5%, AT133-14:19.1% ± 10.1%), and AT133-11 was the weakest agonist (11.9% ± 4.9%) only marginally above the isotype (5.8 ± 4.8%).

When we compared the level of NF-κB transcriptional activity at 6 h with Bmax, on and off rates, and $K_D$, no significant

**Fig. 4 Defining the binding sites of hCD27 mAb. a** The picture on the left depicts the crystal structure of hCD27 as a monomer (extracted from PDB: 5TL5[49]) and the cartoon shows the hCD27 homodimer and the number of amino acid residues in each CRD based on the sequence presented in Supplementary Fig. 4. **b** Representative histograms of the binding of hCD27 mAb and CD70-h1 Fc fusion protein (CD70 h1, CD70) to 293 F cells transfected with WT or domain truncation mutants of hCD27 by flow cytometry. **c** Competition analysis: Human PBMC were incubated with anti-hCD27 m1 (10 µg/ml) or CD70-m1 Fc fusion protein (10 µg/ml) (initial reagent), and then anti-hCD27 h1 (10 µg/ml) or CD70 h1 (10 µg/ml) (detected competing reagent), before binding was detected on CD4[+] T cells by flow cytometry using R-phycoerythrin-conjugated anti-hFc. The heatmap shows the ratio of MFI achieved after PBMC were incubated with the competing reagent, and after PBMC were treated with the respective isotype control. Blue represents blocking/reduced binding and red enhanced binding of the competing reagent. See also Supplementary Fig. 4. **d** Alanine scanning mutagenesis: two consecutive amino acid residues within the hCD27 receptor were replaced with alanine, unless they were cysteine or alanine residues, as exampled. The mutated constructs were then transfected into 293 F cells and hCD27 mAb binding tested by flow cytometry. **e** Representative dot plots for the binding of the hCD27 mAb to the hCD27 construct containing alanine mutations at position W45 and Q47. **f** Heatmap demonstrating binding of hCD27 mAb to pairwise mutated residues. Scales show the ratio of the percentage of hCD27-binding and the percentage of the highest hCD27-binding per mutant. Blue represents no/reduced binding and red mAb binding. Residue pairs predicted to alter protein conformation and impair binding are marked *. **g** Heatmap demonstrating binding of hCD27 mAb to hCD27 with single mutated residues, in CRD1 and CRD2, selected based on the results of the alanine scan with pairwise mutated residues. Scale shows the ratio of the percentage of CD27-binding and the percentage of the highest hCD27-binding per mutant. Blue represents no/reduced binding and red mAb binding.

correlations were observed (Supplementary Fig. 5b, c). However, if hCD27.15 was removed as a visual outlier, a significant association was observed between affinity and activity, with the highest affinity mAb being the most agonistic. A non-significant trend was observed between activity and Bmax (higher Bmax correlated with higher activity). The exception to these observations was hCD27.15.

To investigate the contribution of Fc:FcγR engagement in mAb agonism, the Jurkat NF-κB-GFP reporter assay was repeated in the presence of WT or FcγRIIb-transfected CHO cells and hCD27 h1 mAb (Fig. 6c). Co-incubation with FcγRIIb-CHO cells enhanced the induction of GFP expression by all mAb. The weakest agonists, AT133-11 and AT133-5 showed the greatest increase (6-fold), followed by AT133-2 (4-fold) and varli, AT133-14 and hCD27.15 (3-fold). To better recapitulate the in vivo setting where FcγRIIb expression is lower than that of FcγRIIb-transfected CHO cells (Supplementary Fig. 6), and where other FcγRs are also expressed, we repeated the assay by co-culturing hCD27-transfected Jurkat NF-κB-GFP reporter cells with human PBMC (Fig. 6d). Additionally, we extended the isotypes tested by including h2, and Fc-engineered h1 variants with enhanced affinity for FcγRIIb alone (V11)[22] or both FcγRIIb and FcγRIIa[R131] (SE/LF)[20,22,47]. In cultures without PBMC and thus in the absence of FcγR-mediated cross-linking, hCD27.15 h2 induced the highest increases in %GFP (59.7%). Across all mAb, the h2 isotype produced the highest GFP expression for 4 out of 6 mAb (varli: 34.3%, hCD27.15: 55.6%, AT133-2: 38.4%, AT133-11: 27.8%). However, this was only statistically significant for hCD27.15. Where available, h1 V11 and h1 SE/LF variants did not enhance GFP induction versus native h1 mAb in this setting (without the possibility of further FcγR-mediated cross-linking). In the Jurkat and PBMC co-cultures, addition of PBMC itself did not enhance GFP induction by h1 mAb (Fig. 6d). Increased GFP induction was observed with h2, h1 V11 and h1 SE/LF variants compared to h1. In comparing the level of GFP induced by h2, h1 v11 and h1 SE/LF variants where tested, there was a trend towards h1 V11 and h1 SE/LF producing the highest responses.

We next investigated whether the observed differences in GFP expression also correlated with the strength of downstream signalling. For this experiment, we selected the clinically relevant mAb varli and hCD27.15 (as the strongest agonist) as h1 and h2 isotypes as well as the native ligand CD70 as a positive control. Stimulation of hCD27 Jurkat cells with varli, hCD27.15 and CD70 led to increased p-IκBα expression and corresponding reduction of IκBα expression, indicative of canonical NF-κB activation (Supplementary Fig. 7a, b). Consistent with the Jurkat NF-κB-

reporter cell line data (Fig. 6a, b), the strongest p-IκBα expression was observed with hCD27.15 compared to varli.

An alternate means of determining TNFRSF mAb agonism is by assessing CD8[+] T-cell proliferation in human PBMC cultures in the presence of sub-optimal CD3 stimulation and so we did this in the presence of h1, h2 or h1 V11 isotypes (Fig. 6e–g and Supplementary Fig. 7c). The h1 isotype of all mAb failed to augment CD8[+] T-cell proliferation, and indeed, suppressed it compared to the isotype control (Fig. 6e). In contrast, but consistent with the Jurkat NF-κB-GFP reporter data, h2 isotypes of all mAb were more agonistic. However, this was not significantly different when compared to the isotype control (Fig. 6f).The h1 V11 isotype was highly active and evoked superior CD8[+] T-cell proliferation for 5 of 6 mAb tested (Fig. 6g) versus h2 but this was not statistically significant (hCD27.15 h2 vs hCD27.15 h1 V11: 1.9-fold; AT133-2 h2 vs AT133-2 h1 V11: 1.1-fold; AT133-5 h2 vs AT133-5 h1 V11: 1.3-fold; AT133-11 h2 vs AT133-11 h1 V11: 2.2-fold; AT133-14 h2 vs AT133-14 h1 V11: 1.9-fold). Amongst h1 V11 mAb, varli and AT133-14 were the poorest agonists.

Having established that h2 was more potent in evoking agonism than h1 and in the knowledge that it binds to murine FcγR in a profile similar to m1[19,48] (Supplementary Fig. 2), we next sought to investigate whether the potent agonist hCD27.15 was equally effective with a h2 isotype as a m1 in vivo in hCD27 tg BALB/c mice (Fig. 6h–o). Following dosing on days 0 and 2, we showed that hCD27.15 h2 was able to induce CD8[+] T-cell proliferation to a similar level as hCD27.15 m1 (Fig. 6h). This was also associated with similarly increased Treg numbers (Fig. 6i), elevated CD8/Treg ratio (Fig. 6j), macrophages (Fig. 6k), monocytes (Fig. 6l) and DCs (Fig. 6m). Finally, hCD27.15 h2 was also able to induce upregulation of FcγRIV on both macrophages (Fig. 6n) and monocytes (Fig. 6o) akin to hCD27.15 m1. Together, these investigations indicate that both epitope specificity and isotype contribute to CD27 agonism, and this is apparent in both in vitro human, and in vivo murine settings with mice expressing hCD27.

**Clustering of hCD27 is dependent on epitope and isotype**. We next hypothesised that the improved agonism seen by hCD27.15 compared to other hCD27 mAb of the same isotype, and the superiority of h2, was due to its greater ability to induce clustering of hCD27 at the cell surface. To examine if this was the case, we stimulated hCD27-GFP transfected Jurkat cells with hCD27 h1 mAb (Fig. 7a, c, d). After 6 h, we observed uniform distribution of hCD27-GFP on the cell surface in the isotype-treated cells. A few small clusters were observed on the

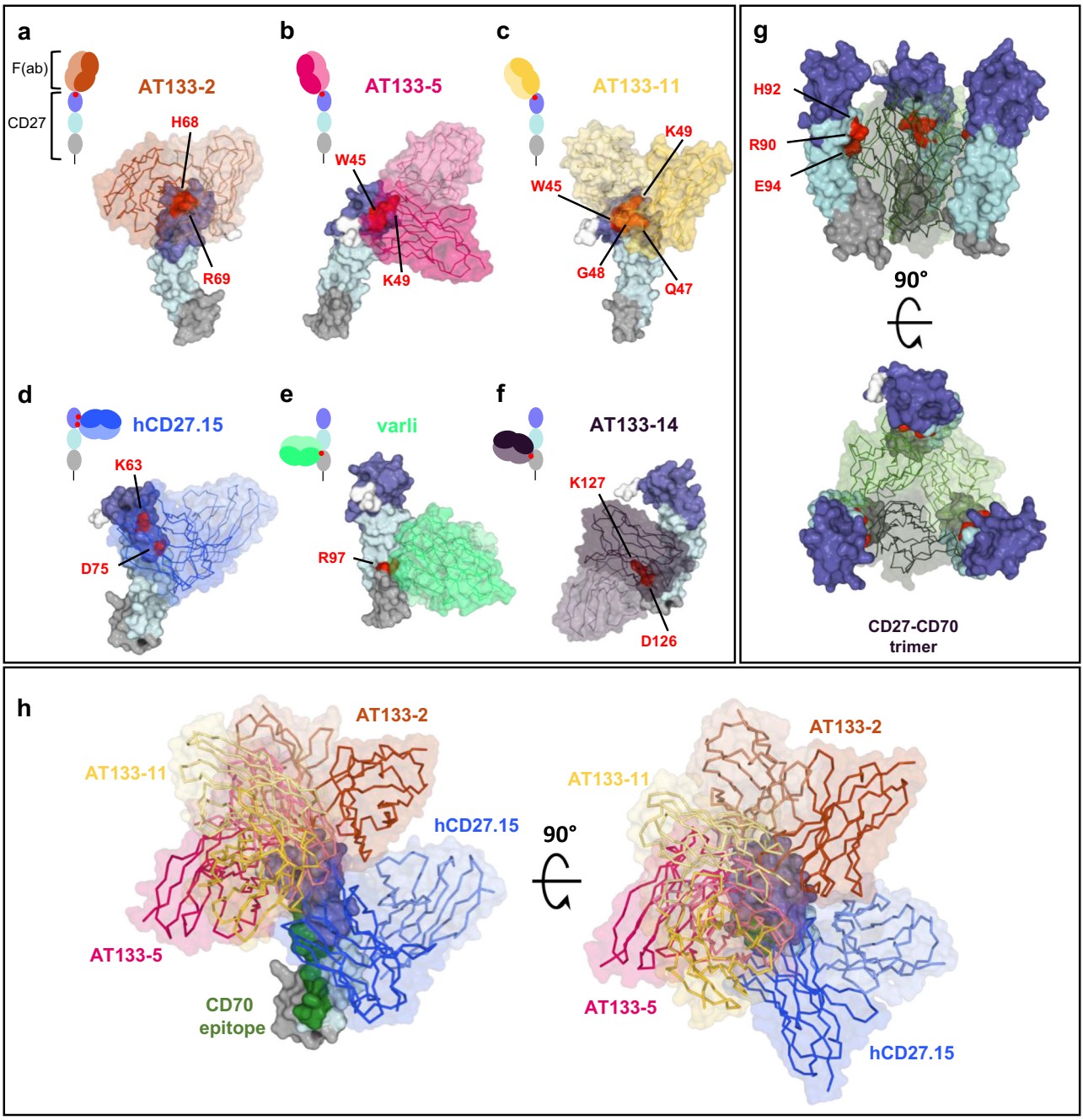

**Fig. 5 Proposed models of binding of the hCD27 mAb determined by in silico docking analysis. a–f** Cartoons: the proposed binding orientations for each of the six hCD27 mAb (**a** AT133-2, **b** AT133-5, **c** AT133-11, **d** hCD27.15, **e** varli, **f** AT133-14) based on docking analysis. Structural models: hCD27 mAb are shown in an opaque surface representation, coloured by domain (as in Fig. 4a, b). hCD27 residues defined as being key to binding (from Fig. 4f, g and Supplementary Table 1), and used as restraints in docking, are coloured and labelled in red. Fv portion of the F(ab) used for docking shown in the respective colour. **g** Proposed model of the hCD27-CD70 trimer (PDB:7KX0) (top image: 'side' view, bottom image: 'top' view). Residues identified from mutagenesis as being key to the interaction, labelled in red, sit at the interface of the CD70 monomers (shades of green). **h** Composite figure of the CRD1-binding Fv, demonstrating opposing binding orientations (left image: 'side' view, right image: 'top' view). Colouring and representation as in **a**. The putative CD70 epitope is highlighted in green on hCD27.

surface after treatment with varli h1, AT133-2 h1, AT133-14 h1 and CD70, but hCD27.15 h1 evoked larger clusters and the highest number of clusters on the cell surface (hCD27.15 vs varli or AT133-14: 3-fold increase, hCD27.15 vs AT133-2: 2-fold increase in numbers of cluster per cell) (Fig. 7d). Next, we employed, h2 isoforms of the same mAb (Fig. 7b–d). Again, isotype-treated cells had uniform distribution of hCD27-GFP.

No difference was observed by isotype switching varli, hCD27.15 or AT133-14 to h2, but increased clustering was observed with AT133-2 h2 versus h1 (1.2-fold) (Fig. 7d). AT133-2 and hCD27.15 are two of the strongest agonists in the NF-κB reporter studies (Fig. 6a–c). This suggests that stronger hCD27 mAb agonism correlates with improved receptor clustering.

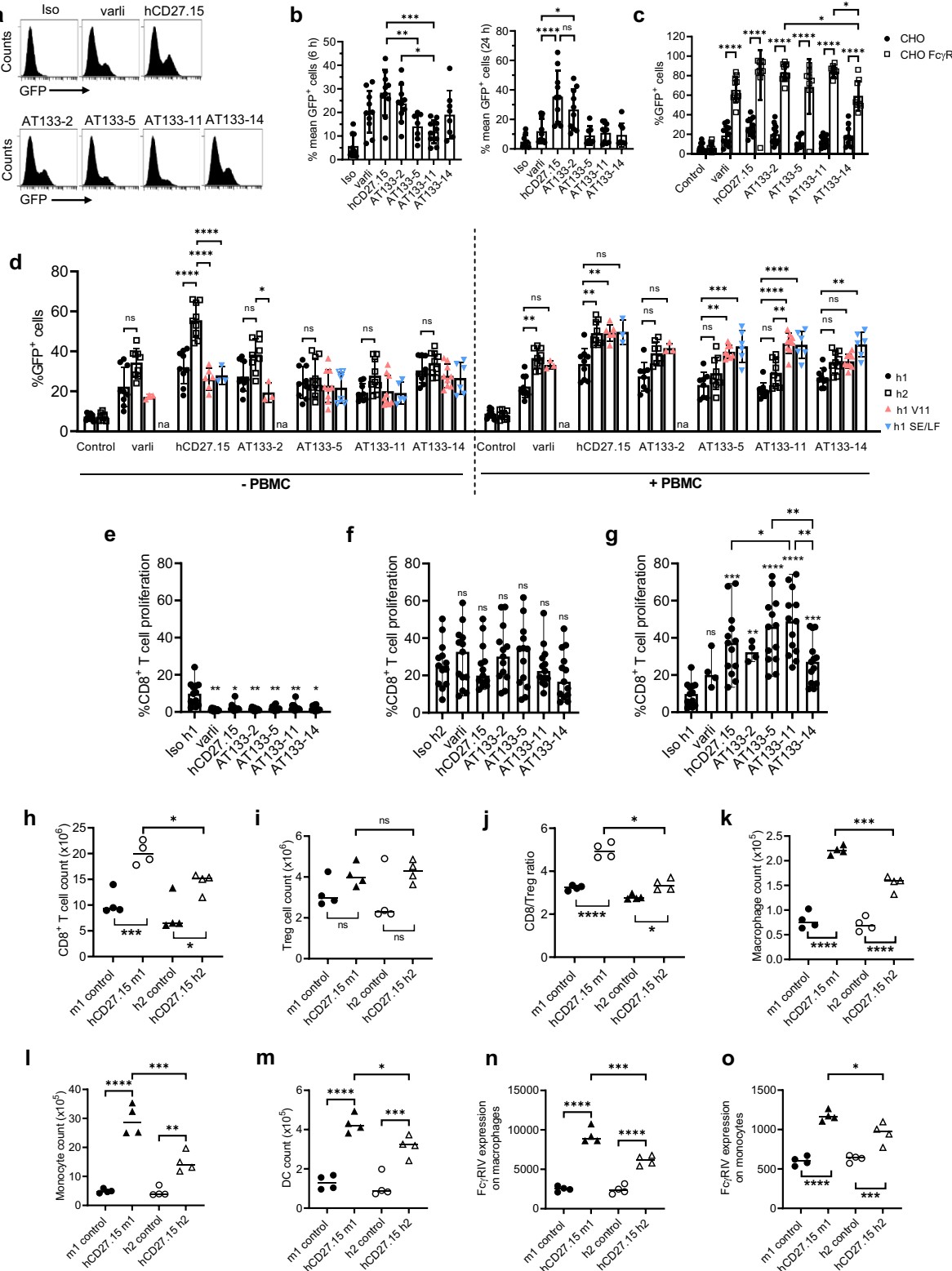

## Discussion

We previously demonstrated that agonistic anti-mCD27 stimulates T cells and NK cells to release CCL3, CCL4, CCL5 and IFNγ, which induced macrophage activation and infiltration into the tumour[16]. In the presence of a direct targeting mAb such as anti-CD20, enhanced ADCP was observed, leading to improved antitumour efficacy across multiple syngeneic murine lymphoma models. In this study, we demonstrate that agonistic anti-mCD27 can also be used to enhance the efficacy of other direct targeting mAb, such as anti-CTLA-4 and anti-CD25 in a colon adenocarcinoma tumour model. Increased intratumoural macrophage infiltration and activation, specifically FcγRIV upregulation, was observed, suggesting that the enhanced Treg depletion by anti-CTLA-4 and anti-CD25 is due to enhanced ADCP. However, the

**Fig. 6 Influence of Fc format on agonism. a, b** NF-κB-GFP hCD27 Jurkat reporter cells were incubated with hCD27 mAb (10 μg/ml) for 6 or 24 h before analysis by flow cytometry. Shown are **a** representative histograms of the GFP-expression 6 h after hCD27 mAb treatment and **b** the graphical display of the GFP-activation at 6 and 24 h. Data representative of n = 8–11 independent experiments. See also Supplementary Fig. 5. **c** NF-κB-GFP hCD27 Jurkat reporter cells were co-cultured with untransfected CHO cells or CHO cells transfected with FcγRIIb for 6 h, and with the isotype control or specified hCD27 mAb (10 μg/ml). GFP activation was assessed by flow cytometry. Data representative of n = 8–11 independent experiments. See also Supplementary Fig. 6. **d** NF-κB-GFP hCD27 Jurkat reporter cells were cultured alone or with PBMC and treated with isotype controls or h1, h2, h1 V9, h1 V11 or h1 SE/LF variants of hCD27 mAb (10 μg/ml) for 6 h. Data compiled from n = 3-9 independent experiments (na: not available). **e–g** PBMC were stimulated with anti-CD3 and **e** h1, **f** h2 or **g** h1 V11 variants of anti-hCD27 (10 μg/ml) for 4 days. Data representative of n = 4-13 independent experiments. Graphs **b–d** show means + SD and **e–g** median with ranges. Underlined statistics in graphs **e–g** show comparisons between mAb and non-underlined statistics comparisons with the respective isotype control. Data were assessed using one-way ANOVA and Tukey's test; *p < 0.05, **p < 0.01, ***p < 0.001, ****p < 0.0001. See also Supplementary Fig. 7. **h–o** hCD27 tg BALB/c mice were treated with hCD27.15 m1 (100 μg) or hCD27.15 h2 (100 μg) or the respective isotype controls on days 0 and 2 and spleens were harvested on day 8. Graphs show absolute numbers of **h** CD8$^+$ T cells, **i** Tregs and **j** CD8/Treg ratio and absolute counts of **k** macrophages, **l** monocytes and **m** DCs and FcγRIV expression on **n** macrophages and **o** monocytes. Graphs are representative of one experiment with n = 4 per group. Shown are medians and one-way ANOVA and Tukey's test were used to assess p-values; *p < 0.05, **p < 0.01, **p < 0.001, ****p < 0.0001.

efficacy of combination therapy is clearly contingent upon the use of a CD27 mAb with the appropriate isotype. Whilst FcγRIIb-binding anti-mCD27 m1 induced CD8$^+$ T-cell proliferation and myeloid cell activation, FcγRIV-binding anti-mCD27 m2a depleted CD8$^+$ T cells and failed to induce myeloid cell activation. Thus, the selection of a sufficiently agonistic CD27 mAb is clearly critical for effective tumour control. With this in mind, we set out to explore what factors might influence anti-CD27-mediated agonism.

Our panel of novel and clinically relevant hCD27 mAb bound differentially across the three extracellular CRD of hCD27. In addition, our alanine scanning mutagenesis and docking analysis data indicate that CD70 binds within CRD2, in line with existing data of its putative binding site[46,49]. All six mAb, except for varli, bound residues not overlapping with those engaged by CD70, however all blocked CD70 binding apart from AT133-2 and hCD27.15. We hypothesise that this is because like CD70, all of the mAb apart from AT133-2 and hCD27.15, bind to the internal surface of a hCD27 homodimer, thus potentially causing steric hindrance to subsequent hCD27:CD70 interaction. One potential factor explaining the inability of hCD27.15 to block the binding of other mAb and CD70 is its high off-rate, which could allow other mAb to bind and out-compete hCD27.15. Our observation that hCD27.15 does not block CD70 binding contradicts existing data by Van Eenennaam et al.[43] The discrepancies in our observations might be due to differences in experimental conditions. Despite using the same mAb and recombinant mouse CD70 fusion protein concentrations, Van Eenennaam et al. employed hCD27 transfected CHO-K1 cells which might express higher levels of hCD27 than PBMC, potentially affecting the binding of hCD27.15.

As well as binding to a different epitope on hCD27, hCD27.15, followed closely by AT133-2, were the strongest agonists in the NF-κB reporter cell assays. When the affinity (by $K_D$) of all the mAb were compared to activity (by NF-κB induction), there was an association of mAb affinity and agonism, but only after hCD27.15 was removed as an outlier from the correlation analysis. hCD27.15 also displayed a greatly different Bmax to the other hCD27 mAb, binding 3 to 4-fold less, suggesting that it binds a unique epitope with high propensity for inherent receptor clustering. Indeed, hCD27.15 was most active across multiple isotypes and in this respect, reminiscent of the strong agonist CD40 mAb CP870,893, which was also isotype agnostic[18,50]. We speculate hCD27.15 may bind across three or more hCD27 homodimer pairs. How a single mAb with only two F(ab) arms can bind across more than two homodimer pairs may be accounted for by its high dissociation rate and its externally-located epitope. We hypothesise that once the mAb binds and

clusters two homodimer pairs (trigger), it is able to rapidly dissociate (release) and bind and bring more homodimers into the cluster. In support of this 'trigger and release' mechanism, others have previously demonstrated that Fas mAb with a high dissociation rate are more agonistic and partial dissociation from the receptor is required for receptor clustering[51]. Our binding models display CD27 mAb binding to monomeric CD27. However, the binding epitopes may be altered by receptor dimerisation secondary to disulphide bonding or through the preligand binding assembly domain (PLAD) formation. To date, the exact arrangement of a CD27 homodimer is still unknown. We speculate that dimerisation of the receptor may reduce the accessibility of mAb binding sites, in particular those internal-facing and membrane-proximal epitopes.

Our observations regarding the influence of mAb isotype on agonism are concordant with those reported for other TNFRSF members (CD40, 4-1BB and OX40)[17,18,27,29]. For CD40, the m1 format, which preferentially engages FcγRIIb is highly agonistic, whereas m2a is inactive[41]. Whilst a similar observation was made for 4-1BB and OX40, the m2a format was not inactive but was able to induce antigen-specific CD8$^+$ T-cell expansion as a consequence of Treg depletion as opposed to direct agonism in order to deliver tumour control[17,29]. Cumulatively, all of these findings including those made by Wasiuk et al.[32] enable us to draw the following conclusions: The therapeutic efficacy of an agonistic immunostimulatory mAb and its mechanism of action is influenced by its isotype and in turn by (1) the abundance of FcγRIIb in the tumour microenvironment, (2) the relative expression of the mAb target on individual cell types, and (3) the reliance of the tumour model on Treg cells, when they express the relevant TNFRSF.

The requirement for CD27 cross-linking is also evident in the human setting, where co-culture with either FcγRIIb-expressing CHO cells, or isotypes with enhanced affinity to FcγRIIb (V11) or FcγRIIa/b (SE/LF) augment NF-κB induction in Jurkat reporter cells as well as CD8$^+$ T-cell proliferation in PBMC cultures compared to h1. In fact, in the PBMC assay, anti-hCD27 h1 treatment reduces CD8$^+$ T-cell proliferation, suggesting possible depletion of CD27$^+$ cells by NK cells in the culture. Further, enhanced clustering of CD27 by h2 compared to h1 mAb was observed in the absence of FcγR-expressing cells, supporting the ability of this isotype to cluster TNFR irrespective of FcγR binding and this is similar to OX40, 4-1BB and CD40 mAb[52]. Based on these findings, hCD27 h2, h1 V11 or SE/LF mAb deliver stronger agonism than h1 counterparts. Compared to B-cell lymphomas, the tumour microenvironment of solid tumours are relatively FcγRIIb-poor, and so here we predict that h2 mAb will be more therapeutically useful than V11 and SE/LF mAb.

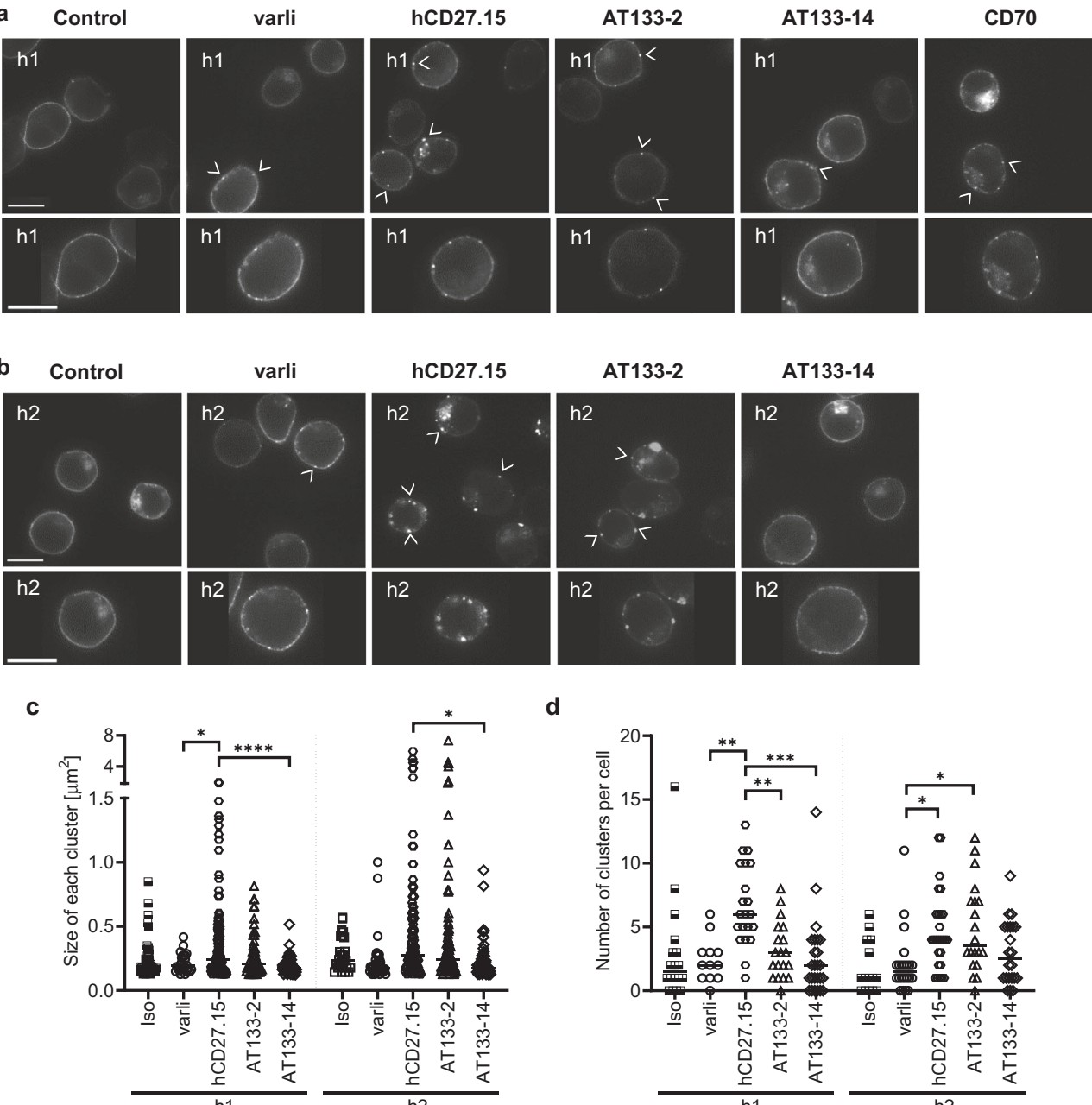

**Fig. 7 Agonistic mAb induce more clustering of hCD27 on the cell surface. a, b** Jurkat cells expressing a hCD27-GFP fusion protein were stimulated with the indicated hCD27 mAb (10 μg/ml) for 6 h to assess receptor clustering by confocal microscopy. Shown are representative images of the 6 h stimulation time point with **a** hCD27 h1 mAb or CD70 h1 or **b** hCD27 h2 mAb with amplification of selected single cells for each mAb. Scale bar represents 10 μm. White arrows indicate surface clusters of hCD27. **c** Size of each cluster upon hCD27 mAb treatment. **d** Quantification of number of clusters per cell. Number of cells counted per condition: Iso h1: 19, varli h1: 12, hCD27.15 h1: 21, AT133-2 h1:19, AT133-14 h1: 25, Iso h2: 14. varli h2: 22, hCD27.15 h2: 27, AT133-2 h2: 20, AT133-14 h2: 22. Graphs show median and data were assessed using one-way ANOVA with Tukey's test; *$p < 0.05$, **$p < 0.01$, ***$p < 0.001$, ****$p < 0.0001$.

Varlilumab is furthest in clinical development and of h1 isotype. Its clinical efficacy is modest and definitive evidence of agonism, or indeed, understanding behind the mechanism of action, is lacking. Our data suggests that employment of a h1 isotype may contribute to the lack of observed agonism, and provides a guide towards engineering a more potent and more clinically efficacious CD27 mAb.

## Methods

**Mice**. BALB/c mice were obtained from Charles River Laboratories, maintained in local facilities and the stock refreshed every six generations. hCD27 tg BALB/c

mice[44] and FcγRIV knock out (FcγRIV[-/-]) BALB/c mice[40] were bred and maintained in local facilities. All mice were fed regular chow, had water freely and were maintained in a conventional facility. All experiments were conducted with age-matched (8–12 week old) female mice. Animals were randomly assigned to experimental groups and housed together under the same conditions. All experiments were conducted according to the UK Home Office license guidelines and in accordance with the Animals (Scientific Procedures) Act 1986 under the Procedure Project Licences P81E129B7 and P4D9C89EA and approved by the University of Southampton Ethical Committee.

**Cell Lines**. The BCL₁ B-cell lymphoma cell line[53] originates from female mice (age of source mice unknown) and was maintained by passaging in BALB/c mice[54].

CT26 colon carcinoma cells[55] were obtained from ATCC and cultured in Roswell Park Memorial Institute (RPMI) 1640 medium supplemented with 10% foetal calf serum (FCS), 2 mM L-glutamine (Life Technologies), 1 mM pyruvate (Life Technologies) and 100 U/ml penicillin/100 μg/ml-streptomycin (Life Technologies) (complete RPMI (cRPMI) medium). hCD27 transfected Jurkat cells with (System Biosciences) or without NF-κB-GFP reporter were generated by lipofection and maintained in cRPMI medium and puromycin. hCD27/GFP transfected Jurkat cells (ATCC® TIB-152) used in microscopy were generated using Amaxa Cell line Nucleofector Kit V according to the manufacturer's instructions and cultured in cRPMI medium and geneticin. Wild-type CHO cells (ECACC 85051005) were cultured in Glasgow's modified Eagle's medium (First Link (UK) Ltd.) supplemented with 5% FCS, 2 mM L-glutamine, 1 mM pyruvate and 100 U/ml penicillin-streptomycin. CHO cells stably transfected with FcγRIIb by lipofection using Gene Porter transfection reagent were a kind gift of Dr R. Oldham[56]. Cells were cultured in cRPMI medium with geneticin (Life Technologies). 293 F cells were cultured in Freestyle293 media. All cell lines were maintained at 37 °C and 5% CO₂ atmosphere in a humidified incubator.

**Human samples**. PBMC were obtained from fresh leucocyte cones received from anonymised, adult healthy donors through the National Health Service Blood and Transplant service. The use of human leucocyte blood cones was approved by the East of Scotland Research Ethics Service (REC reference [16]/ES/0048), in accordance with the Declaration of Helsinki. PBMC were isolated by density gradient centrifugation using Lymphoprep™ and cultured in cRPMI.

**Tumour models**. For CT26 tumour experiments, BALB/c mice were subcutaneously injected with $5 \times 10^5$ CT26 colon carcinoma cells at day 0 followed by intraperitoneal administration of 200 μg anti-mCTLA-4 m2b (9D9) on day 10, 13, 16 and 19 or 100 μg anti-mCD27 m1 (AT124-1) on day 11, 13, 16 and 18 or the combination.

For BCL₁ tumour experiments, mice were intravenously injected with $10^4$ BCL₁ tumour cells on day 0. For the anti-CD20/anti-mCD27 combination study, WT BALB/c mice received 200 μg anti-CD20 m2a (18B12) on day 4 or 50 μg anti-mCD27 m1 or m2a (AT124-1) on days 5-8 or PBS (control) or the combination via intraperitoneal injection.

For the investigation of mAb isotype on anti-tumour activity, WT BALB/c mice were intraperitoneally injected with either 50 μg anti-mCD27 m1 or m2a (AT124-1) or PBS (control) on days 5-8. Spleens for the assessment of immune cell subsets were harvested on days 9 and 13, respectively. To explore the role of FcγRIV, WT BALB/c or FcγRIV$^{-/-}$ mice received PBS or 50 μg anti-mCD27 r2a (AT124-1) on days 5-8.

hCD27 tg BALB/c mice were treated with hCD27.15 m1 or h2 (100 μg, respectively) on day 0 and day 2 and spleens were harvested on day 8.

**Antibodies**. Anti-mCTLA4 m2b (9D9) was purchased from BioXCell. 18B12 (anti-mCD20)[57], varli (anti-hCD27)[30,31,58,59] and hCD27.15 (anti-hCD27)[43] were produced in-house from published patented sequences. hCD27 mAb AT133-2, AT133-5, AT133-11 and AT133-14 were generated in house using standard hybridoma technology after immunisation of mice with recombinant human CD27/TNFRSF7 Fc chimera protein (R&D Systems). Spleens from immunised mice were fused with NS-1 myeloma cells and plates screened by ELISA and flow cytometry. Transient antibody production was performed using ExpiFectamine CHO Transfection Kit (Gibco) and antibodies were purified using affinity chromatography. All antibodies were tested for endotoxin levels (<5 EU/mg) and regularly checked for aggregation by HPLC and de-aggregated (if aggregation >1%) by size exclusion chromatography.

**Surface plasmon resonance analysis**. Anti-His-antibodies (GE Healthcare Life Sciences) were immobilised onto a CM5-SPR-chip. Subsequently, His-tagged hCD27 (produced in-house using the His Capture Kit, GE Healthcare Life Sciences) was captured onto the chip at 25 °C. For determination of IgG binding affinities, mAb were diluted to 250, 50, 10, 2 and 0.4 nM in HBS-EP running buffer (GE Healthcare) and the interaction was analysed using a Biacore T200 system. For the determination of binding kinetics, a 1:1 binding fitting curve was applied using Biacore T200 Evaluation software and GraphPad Prism 9.

**Flow cytometry**. Cells were suspended in staining buffer (PBS with 1% w/v BSA, 0.1% w/v sodium azide) to get single cell suspensions. For flow cytometric analysis of murine tissue samples, FcγR-block (2.4G2) was added at 20 μg/ml for 10 min at RT and, without washing, cells were stained for 15 min at RT with antibodies specific for the markers of interest. Where markers of interest were FcγR, no FcγR-block was used. Subsequently, red blood cell lysis reagent (Invitrogen) was added for 2–3 min, cells washed twice more and fixed in 2% paraformaldehyde for flow cytometry. For intracellular staining, cells were fixed and permeabilised using Foxp3 Transcription Factor Fixation/Permeabilisation Concentrate (eBioscience) for 30 min at RT. After two washes, cells were stained for 20 min, washed twice again and analysed by flow cytometry. For enumeration and analysis of myeloid cells, tissue was digested using Liberase TL (Sigma Aldrich) according to the manufacturer's protocol. To do so, tissue was chopped into small pieces, treated

with Liberase TL for 15 min at 37 °C and mashed into a single cell suspension. All flow cytometric analysis were performed using a FACSCalibur or FACSCanto (BD Biosciences, respectively) and analysed using Cytobank[60].

**Binding analysis of hCD27 mAb**. Isolated human PBMC were plated at $1 \times 10^7$ cells/ml in a flat-bottom 96 well plate and incubated with hCD27 mAb at indicated concentrations for 30 min at 4 °C. Subsequently PBMC were washed and binding was determined on CD3⁺CD4⁺ T cells using a secondary R-phycoerythrin-conjugated anti-hFc.

**CD8⁺ T-cell proliferation**. PBMC were isolated as described and labelled with CFSE (eBioscience) at 2 μM for 10 min at RT. The reaction was quenched and washed with cRPMI media. Cells were cultured in a flat-bottom 24-well plate at $1 \times 10^7$ cells/ml in cRPMI for 24 h. PBMC were harvested and plated in a round bottom 96-well plate at $1 \times 10^6$ cells/ml in cRPMI medium and stimulated with 0.5 ng/ml soluble CD3 mAb (OKT3) as well as hCD27 mAb and respective irrelevant isotype controls at 10 μg/ml for 4 days. Proliferation of CD8⁺ T cells was assessed by flow cytometry, measured through a reduction in CFSE staining.

**NF-κB reporter assays for the investigation of the agonistic activity of hCD27 mAb**. 24 h prior to the assay, cells were transferred to selection-free medium. NF-κB-GFP hCD27 Jurkat cells were cultured in a flat-bottom 96-well plate at a concentration of $1.5 \times 10^6$ cells/ml in a final volume of 200 μl. hCD27 mAb were added in solution at a concentration of 10 μg/ml and cells were subsequently incubated for 6 h or 24 h. For determination of the influence of FcγR cross-linking, FcγRIIb-transfected CHO cells were transferred to selection-free media 24 h prior to the assay. The cells were placed in a flat bottom 96-well plate at a concentration of $1 \times 10^5$ cells/ml in a final volume of 200 μl and incubated overnight. NF-κB-GFP hCD27 Jurkat cells were co-cultured at $1.5 \times 10^6$ cells/ml in 200 μl and stimulated with hCD27 mAb at 10 μg/ml for 6 h. For the co-culture of NF-κB-GFP hCD27 Jurkat cells with PBMC, frozen PBMC were thawed in cRPMI and plated in a flat-bottom 96-well plate at a concentration of $1 \times 10^5$ cells/ml in a final volume of 100 μl. NF-κB-GFP hCD27 Jurkat cells were subsequently added at a concentration of $2 \times 10^4$ cells/ml and hCD27 mAb were added at a concentration of 10 μg/ml and incubated for 6 h. NF-κB transcriptional activation was assessed by flow cytometry as an increase in GFP, detected in the FL1 channel.

**Domain-mapping using hCD27 truncation mutants**. hCD27 CRD1, CRD1+2, CRD2+3 and CRD3 truncation mutants were generated in-house using site directed mutagenesis and PCR cloning. Constructs were transiently transfected into 293 F cells using FreeStyle MAX Reagent (ThermoFisher Scientific) and incubated for 24 h. For flow cytometric analysis, cells were incubated with hCD27 mAb or soluble CD70-h1 Fc fusion protein[61] at 10 μg/ml for 30 min at 4 °C. After washing, the cells were stained with a secondary R-phycoerythrin-conjugated anti-hFc for 30 min at 4 °C and binding was analysed using flow cytometry.

**Cross-blocking competition assay for hCD27 mAb**. Human PBMC were isolated as described and $1 \times 10^6$ cells were incubated with 10 μg/ml hCD27 m1 mAb or CD70-m1 Fc fusion protein for 30 min at 4 °C. Without washing, secondary hCD27 h1 mAb or CD70-h1 fusion protein was added at 10 μg/ml and incubated for 30 min at 4 °C. After washing the PBMC were stained with a R-phycoerythrin-conjugated anti-hFc as well as CD3 and CD4 mAb for 30 min at 4 °C and binding was assessed on CD3⁺CD4⁺ T cells using flow cytometry. For analysis, the ratio of PBMC stained with hCD27 mAb and PBMC stained with the relevant isotype controls was calculated.

**Alanine scanning mutagenesis**. For detailed epitope mapping, mutated hCD27 DNA constructs were generated by GenScript. The constructs were composed of the extracellular, transmembrane and intracellular region of hCD27 as well as a CD27 leader sequence and a rituximab epitope tag proximal to the hCD27 extracellular region[62]. Mutants were generated by replacing two consecutive amino acids or single amino acids to alanine, except when alanines were already present or if the residue was a cysteine. 293 F cells were transiently transfected with the generated hCD27 constructs using FreeStyle MAX Reagent (ThermoFisher Scientific) and incubated for 24 h. For flow cytometric analysis, cells were incubated with hCD27 mAb, soluble CD70-h1 Fc fusion protein and according isotype controls at 10 μg/ml for 30 min at 4 °C. After washing, the cells were stained with a secondary R-phycoerythrin-conjugated anti-hFc for 30 min at 4 °C. To confirm protein expression of the constructs on the cell surface, cells were stained with an anti-rituximab FITC mAb at 10 μg/ml for 30 min at 4 °C. Binding was determined using flow cytometry. To normalise for different transfection levels, binding was calculated as a ratio of %CD27⁺ cells and the highest % of CD27-expression per mutant.

**Docking**. To investigate the potential differences in binding location and conformation of the F(ab) of interest, information-driven F(ab)-CD27 docking was performed. Homology models of the Fv domains of the 6 F(ab) of interest were generated[63]. The CD27 structure was then extracted from PDB:5TL5[49]. Docking

was performed using HADDOCK2.4 with restraints imposed from the site directed mutagenesis data[64,65]. Fv hypervariable loops were defined as active residues, while the CD27 residues identified as critical to binding were defined as passive residues (Supplementary Table 1). All sampling parameters were left as default, with rigid body sampling undergoing 1000 steps, semi-flexible refinement 200 steps and water refinement 200 steps. Optimal models were selected based on their HADDOCK score (Supplementary Table 2).

**Confocal microscopy**. hCD27/GFP Jurkat cells were cultured in a flat-bottom 96-well plate in solution at $1.5 \times 10^6$ cells/ml and stimulated with 10 µg/ml hCD27 mAb for 6 h at 37 °C. After the incubation, cells were washed with PBS and fixed with 4% paraformaldehyde for 10 min at RT. After fixation, cells were washed with PBS and transferred to a poly-L-lysine covered µ-Slide 8 well microscopy chamber slide (Ibidi) and centrifuged at 250 g for 10 min to gently attach the cells to the chamber slide and stored for up to 24 h in darkness in PBS at 4 °C prior to analysis. Cells were acquired in confocal mode (100x objective lens) using an ONI Nanoimager (ONI Oxford) and analysed using NimOS 1.18 and ImageJ software. For cluster enumeration, fluorescent puncti ≥ 0.126 µm$^2$ were defined as a cluster.

**Western blot**. For the production of cell lysates, hCD27 Jurkat cells were cultured in a flat bottom 96-well plate at $1.5 \times 10^6$ cells/ml and stimulated with the hCD27 mAb for 5 min at 37 °C and 5% $CO_2$ in a humidified incubator. Stimulation was stopped by transferring the cells to ice and centrifugation at 400 g for 5 min at 4 °C. Cell pellets were resuspended in radioimmunoprecipitation buffer[66] and incubated on ice for 30 min. Cell lysates were stored at −20 °C. A Bradford assay was performed to determine protein concentrations and 15 µg of protein were loaded for the following western blot. Subsequently, SDS PAGE and protein transfer onto nitrocellulose or polyvinylidene difluoride membranes (Invitrogen) were performed. The membrane was blocked with 5% milk in Tris-buffered saline with 0.1% Tween 20 (TBS-T) and subsequently stained with the primary antibody over night at 4 °C followed by washing in TBS-T and application of the secondary mAb for 2 h at RT. To visualise bands, membranes were incubated with ECL substrate (Abcam) and the membrane was imaged using a Biospectrum AC Imaging system and analysed using ImageJ software.

**Statistics and reproducibility**. Flow cytometry data analysis was conducted using Cytobank Software Version 7.2. All other data analysis was performed using GraphPad Prism 9 software. One-way ANOVA followed by Tukey's multiple comparisons test were performed to assess differences between two or more groups. For paired comparisons, Student's t-test (un-paired, two-tailed) was used. To assess survival differences in immunotherapy experiments, Kaplan-Meier curves were generated and Log-rank tests were utilised for the statistical analysis.Not significant (ns) ≥ 0.05, *$p < 0.05$, **$p < 0.01$, ***$p < 0.001$, ****$p < 0.0001$. Reproducibility including technical replicates and independent biological experiments are stated in each figure legend.

**Reporting summary**. Further information on research design is available in the Nature Research Reporting Summary linked to this article.

## Data availability

All data supporting the findings of this study are available within the published article and its supplementary information files. Source data are provided in Supplementary Data 1. All other data are available from the corresponding author upon reasonable request.

## Material availability

hCD27 mAb generated in this study will be made available on reasonable request, but we may require a payment and/or Material Transfer Agreement if there is potential for commercial application. There are restrictions to the availability of anti-CD32b (6G11) due to a Materials Trasfer Agreement being required with BioInvent International.

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

## Acknowledgements

We thank Alison L. Tutt for the generation of the CD27 mAb. We would also like to thank the Biomedical Research Facility and pre-clinical unit for animal husbandry. The microscopy work was made possible through the generous funding of an ONI Nanoimager by the Mark Benevolent Fund. This work was funded by a UK Medical Research Council Industrial Collaborative Awards in Science and Engineering (iCASE) studentship, Celldex Therapeutics, Cancer Research UK Advanced Clinician Scientist Fellowship to S.H.L. (A27179) and Cancer Research UK Centre funding (A27452).

## Author contributions

Conceptualisation, S.H.L.; Methodology, S.H.L. and F.H.; Investigation: F.H., A.H.T., H.F., H.T.C.C., M.J.E.M, C.A.P., T.I. and C.I.M.; Resources, O.D.; Formal analysis, F.H., A.H.T and S.H.L; Writing – original draft, F.H.; Writing - Review & Editing, S.H.L., S.A.B., M.S.C., D.A., I.T. and T.K; Supervision, S.H.L. All authors reviewed and approved the manuscript.

## Competing interests

S.H.L. is a co-inventor on a patent application filed (JDM84560P.GBA), receives research funding from and has acted as a consultant to Celldex Therapeutics. F.H. receives research funding from Celldex Therapeutics. M.S.C. is also a co-inventor on patent application filed (JDM84560P.GBA). D.A. and T.K. are employees and shareholders of Celldex Therapeutics. The remaining authors declare no competing interests.
