## [Peer Review File · Communications Biology]

Reviewers' comments:

Reviewer #1 (Remarks to the Author):

The authors present their work on agonistic CD27 antibodies, showing the effects of binding affinity, binding kinetics, epitope, and IgG subclass on receptor clustering and agonism. Using CT26-bearing mice, it is demonstrated that addition of CD27 mAb to CTLA4 mAb therapy increases the CD8/Treg ratio and overall survival. The isotype of CD27 mAb seems to play a key role, with the mouse IgG1 subtype leading to better survival than the IgG2a subtype. The authors also evaluate the binding affinity for a panel of CD27 mAbs, and carry out several experiments to identify their epitopes, including binding to CD27 subdomains, competitive blocking, scanning mutagenesis, and computational docking. Finally, it is shown that the different mAbs have distinct abilities to cluster and activate CD27 signaling, with potential correlations to epitope or binding kinetics. Overall, the article provides an interesting analysis of the determinants of CD27 mAb agonism, and more generally explores the properties of mAbs that could make them better modulators of TNFRSF signaling.

Major points:

- A crystal structure of the CD27:CD70 complex was recently published ([https://www.jbc.org/article/S0021-9258\(21\)00905-4/fulltext](https://www.jbc.org/article/S0021-9258(21)00905-4/fulltext)). I think the structure is mostly in line with the authors' modeling predictions (figure 5g), but any similarities/differences should be described in the discussion.
- Would it be possible to make amino acid numbering consistent between Figures 4a and 4f/g? It seems like the numbering in 4f/g is shifted to larger numbers, which makes it difficult to compare these positions to the structure in 4a.
- In figure 4c, MK-5890 is unable to block the binding of other mAbs and CD70, and the authors mention in the discussion that this result is inconsistent with a previous publication showing that MK-5890 blocks CD70. Based on the observations that MK-5890 has the lowest MFI response in figure 3b and has low affinity, is it possible that the concentration used is not sufficient to completely occupy all CD27 receptors, despite the apparent saturation observed in figure 3b? The KD of MK-5890 is 30 nM, which indicates that the antibody would only be 50% bound at 30 nM = 4.5 ug/ml. The competition experiment only uses 10 ug/ml. Plus, MK-5890 has the highest off-rate, which could allow other antibodies to out-compete it in real time and stay attached. It would be useful to discuss these potential limitations when the discrepancy with previous reports is brought up.
- Can the authors speculate how the dimerization of CD27 on cells would impact binding epitope compared to the monomeric form shown in Figure 5, if at all? For example, would CD27 dimerization through a certain interface affect binding of antibodies to that epitope?
- In the "Antibodies" methods section, can you please provide more information on how the antibodies were expressed/purified? For experiments involving clustering and agonism, it's possible that aggregation could impact results, so it would be important to use SEC to remove aggregate, if necessary.
- In Supplemental Figure 3a, the gating strategy for CD4+/CD8+ T cells is shown, but it doesn't seem like CD8 was explicitly stained.
- In Supplemental Figure 5, correlations are found after removing the datapoint for MK-5890 which is deemed an outlier. With only 6 mAbs, it seems statistically unwarranted to remove one datapoint in order to generate statistically significant correlations. I agree it is a tantalizing result that activity seems to be correlated with KD, but inclusion of all datapoints eliminates this trend. Removing a different datapoint could yield a totally different result. For example, removing the mAb on the right of the Activity vs. On rate plot looks like it would yield a decent positive correlation. I think this issue could be easily remedied by being a bit more conservative in the wording in the results and discussion.
 - o For example, on line 338, it could be stated similar to the following:
When we compared the level of NK-kB transcriptional activity at 6 h with Bmax, on and off rates, and KD, no significant correlations were observed. However, if MK-5890 was removed as a visual outlier, a significant association was observed between affinity and activity, with the highest mAb being the most agonistic.
 - o Similarly, on line 473:
When the affinity (by KD) of all the mAb were compared to activity (by NK-kB induction), there

was an association of mAb affinity and agonism, but only if MK-4890 was removed from the correlation analysis.

Minor points:

- On line 89, I think maybe "determinates" should be "determinants."
- On line 119-120, the phrase "using agonistic anti-CD27 to enhance depleting mAb" is a bit confusing. Perhaps it could read, "using agonistic anti-CD27 to enhance the effects of Treg-depleting mAb."
- In, for example, figure 1, I was confused how tumor could be harvested on day 20 when the experiment continued much longer. Perhaps it was a different group of mice? Clarification would be helpful.
- On line 164, it mentions "minimal change was observed in monocyte numbers." However, it looks like there was a significant decrease in monocytes on D13.
- In the paragraph beginning on line 197, it would be worthwhile to mention that the KD measured in this experiment is in fact avidity; due to bivalent binding, the KD may deviate from the true affinity of a single mAb:CD27 interaction.
- For figure 3b, a semilog plot (log x-axis) could help to visualize the hyperbolic data.
- On line 239, it could also be mentioned that the CRD3-binding AT133-14 also blocked binding of CD70.
- On line 250, when pointing out that MK-5890 epitope is located at CRD1 and 2, the residues I110 and T111 on CRD2 could also be mentioned, since both the listed residues are in CRD1.
- On line 274, the sentence "The heatmap shows the ratio of PBMC incubated with the competing reagent, and PBMC treated with the respective isotype control" is unclear. Perhaps it could read, "The heatmap shows the ratio of MFI achieved after PBMC was incubated with the competing reagent, and after PBMC was treated with the respective isotype control."
- Throughout the paper Fab and F(ab) seem to be used interchangeably. It would be clearer to use one version throughout.
- In the legend for Figure 5h, I think it should say "CRD1-binding Fv" rather than "CRD1-binding F(ab)."
- On line 357, the increase in GFP expression for h2 "was not statistically significant." But in figure 6d, it looks like the increase is significant for MK-5890.
- For figure 6, the term "Fc specificity" in the title is unclear to me. Perhaps it could be "Fc subtype", "Fc format", or simply "Fc"? However I understand if this is a matter of personal taste.
- On line 410 there is a typo, where the word "show" is repeated.
- On line 428, I think "AT133-14" should be "AT133-2".
- On lines 424 and 429, when Fig. 7a and 7c are cited, I think 7d should also be cited.
- On line 512, "poorly understood" could be replaced with "lacking", to avoid repetition.
- I didn't see methods for the CT26 mice experiments in the "Tumor Models" section.
- The supplementary figures specify which main figures they are related too, but sometimes this seems too specific. For example, Supplemental Figure 1 says it is related to Figure 1a/b, but it seems to be important for Figure 1 more generally.
- In Supplemental Figure 7, it would be interesting to include CD70 in panel B for comparison, although tubulin intensity for CD70 in panel A seems weaker than for the other samples for some reason.

Reviewer #2 (Remarks to the Author):

Lim and coworkers investigated the efficacy of agonistic CD27 monoclonal antibodies. They generated a set of new antibodies and compared them with two antibodies already in the clinic. These antibodies were characterized using state of the art methods. Epitope mapping was performed with truncated receptor variants and also with receptor mutant pairs. Importantly, the authors speculated that poor epitope-dependent agonism could be counteracted by Fc engineering, particularly by using isotypes promoting receptor clustering. Their data indicate that FcγRIIb-binding anti-mCD27 m1 induces CD8+ T-cell proliferation and myeloid cell activation. The authors conclude that engagement of activatory FcγR may be detrimental to therapy with ramifications for further development of anti CD27 therapeutic antibodies. The authors provide a very remarkable wealth of data. The paper certainly merits publication. The only very minor point I have is as

follows: P10 li 202 Precision of KD values is overestimated, please round to one digit (e.g. 30.77 =30.8).

Reviewer #3 (Remarks to the Author):

The authors present an thorough and very detailed study how antibody isotype, epitope specificity and affinity determine the agonistic activity of anti-CD27 antibodies. A key finding is that the IgG2 isotype is able to mediate CD27 receptor clustering independent of FcR binding. The experiments are very well performed providing combining in vitro and in vivo analyses supporting their conclusions. Very well written.

I only have some minor comments.

Please explain the treatment scheme shown in Fig.1. Why were molecules used as single treatments and then in the middle as combination treatment. Why were not all treatments combination treatments?

When analysing T-regs in Fig. 1, it says tumors were harvested on day 20. Are these the same animals as shown in Fig.1c? How could some animals survive beyond day 20 when tumors were harvested?

Would it be possible to include epitope data in Suppl. Fig. 4?

We thank the reviewers for the time taken to review our manuscript, and for noting that the manuscript is interesting, provides a 'remarkable wealth of data' and also 'very well written'. Our point-by-point response is enclosed in the table below.

In addition to this, we have also relabelled the antibody named "MK-5890" to "hCD27.15". This mirrors the clone name on the US patent (US9527916B2 by Van Eenennaam et al.) from which we derived the antibody sequence and avoids any confusion with respect to the antibody used in our studies.

Finally, we have also increased the sample size for Figure 6d and 6e-f and updated the figure accordingly (Fig. 6d: n=3-6 to n=3-9; Fig. 6e-f: n=4-5 to n=4-13). Accordingly, the main text (Lines 380-385) and figure legends (Lines 409-413) were adjusted.

Reviewer #1:

Major points:

1. A crystal structure of the CD27:CD70 complex was recently published ([https://www.jbc.org/article/S0021-9258\(21\)00905-4/fulltext](https://www.jbc.org/article/S0021-9258(21)00905-4/fulltext)). I think the structure is mostly in line with the authors' modeling predictions (figure 5g), but any similarities/differences should be described in the discussion.
2. Would it be possible to make amino acid numbering consistent between Figures 4a and 4f/g? It seems like the numbering in 4f/g is shifted to larger numbers, which makes it difficult to compare these positions to the structure in 4a.
3. In figure 4c, MK-5890 is unable to block the binding of other mAbs and CD70, and the authors mention in the discussion that this result is inconsistent with a previous publication showing that MK-5890 blocks CD70. Based on the observations that MK-5890 has the lowest MFI response in figure 3b and has low affinity, is it possible that the concentration used is not sufficient to completely occupy all CD27 receptors, despite the apparent saturation observed in figure 3b? The KD of MK-5890 is 30 nM, which indicates that the antibody would only be 50% bound at 30 nM = 4.5 ug/ml. The competition experiment only uses 10 ug/ml. Plus, MK-5890 has the highest off-rate, which could allow other antibodies to out-compete it in real time and stay attached. It would be useful to discuss these potential limitations when the discrepancy with previous reports is brought up.
4. Can the authors speculate how the dimerization of CD27 on cells would impact binding epitope compared to the monomeric form shown in Figure 5, if at all? For example, would CD27 dimerization through a certain interface affect binding of antibodies to that epitope?
5. In the "Antibodies" methods section, can you please provide more information on how the antibodies were expressed/purified? For experiments involving clustering and agonism, it's possible that aggregation could impact results, so it would be important to use SEC to remove aggregate, if necessary.
6. In Supplemental Figure 3a, the gating strategy for CD4+/CD8+ T cells is shown, but it doesn't seem like CD8 was explicitly stained.
7. In Supplemental Figure 5, correlations are found after removing the datapoint for MK-5890 which is deemed an outlier. With only 6 mAbs, it seems statistically unwarranted to remove one datapoint in order to generate statistically significant correlations. I agree it is a tantalizing result that activity seems to be correlated with KD, but inclusion of all datapoints eliminates this trend. Removing a different datapoint could yield a totally

different result. For example, removing the mAb on the right of the Activity vs. On rate plot looks like it would yield a decent positive correlation. I think this issue could be easily remedied by being a bit more conservative in the wording in the results and discussion.

- For example, on line 338, it could be stated similar to the following:

When we compared the level of NK-kB transcriptional activity at 6 h with Bmax, on and off rates, and KD, no significant correlations were observed. However, if MK-5890 was removed as a visual outlier, a significant association was observed between affinity and activity, with the highest mAb being the most agonistic.

- Similarly, on line 473:

When the affinity (by KD) of all the mAb were compared to activity (by NK-kB induction), there was an association of mAb affinity and agonism, but only if MK-4890 was removed from the correlation analysis.

Minor points:

1. On line 89, I think maybe “determinates” should be “determinants.”
2. On line 119-120, the phrase “using agonistic anti-CD27 to enhance depleting mAb” is a bit confusing. Perhaps it could read, “using agonistic anti-CD27 to enhance the effects of Treg-depleting mAb.”
3. In, for example, figure 1, I was confused how tumor could be harvested on day 20 when the experiment continued much longer. Perhaps it was a different group of mice? Clarification would be helpful.
4. On line 164, it mentions “minimal change was observed in monocyte numbers.” However, it looks like there was a significant decrease in monocytes on D13.
5. In the paragraph beginning on line 197, it would be worthwhile to mention that the KD measured in this experiment is in fact avidity; due to bivalent binding, the KD may deviate from the true affinity of a single mAb:CD27 interaction.
6. For figure 3b, a semilog plot (log x-axis) could help to visualize the hyperbolic data.
7. On line 239, it could also be mentioned that the CRD3-binding AT133-14 also blocked binding of CD70.
8. On line 250, when pointing out that MK-5890 epitope is located at CRD1 and 2, the residues I110 and T111 on CRD2 could also be mentioned, since both the listed residues are in CRD1.
9. On line 274, the sentence “The heatmap shows the ratio of PBMC incubated with the competing reagent, and PBMC treated with the respective isotype control” is unclear. Perhaps it could read, “The heatmap shows the ratio of MFI achieved after PBMC was incubated with the competing reagent, and after PBMC was treated with the respective isotype control.”

10. Throughout the paper Fab and F(ab) seem to be used interchangeably. It would be clearer to use one version throughout.
11. In the legend for Figure 5h, I think it should say “CRD1-binding Fv” rather than “CRD1-binding F(ab).”
12. On line 357, the increase in GFP expression for h2 “was not statistically significant.” But in figure 6d, it looks like the increase is significant for MK-5890.
13. For figure 6, the term “Fc specificity” in the title is unclear to me. Perhaps it could be “Fc subtype”, “Fc format”, or simply “Fc”? However I understand if this is a matter of personal taste.
14. On line 410 there is a typo, where the word “show” is repeated.
15. On line 428, I think “AT133-14” should be “AT133-2”.
16. On lines 424 and 429, when Fig. 7a and 7c are cited, I think 7d should also be cited.
17. On line 512, “poorly understood” could be replaced with “lacking”, to avoid repetition.
18. I didn’t see methods for the CT26 mice experiments in the “Tumor Models” section.
19. The supplementary figures specify which main figures they are related too, but sometimes this seems too specific. For example, Supplemental Figure 1 says it is related to Figure 1a/b, but it seems to be important for Figure 1 more generally.
20. In Supplemental Figure 7, it would be interesting to include CD70 in panel B for comparison, although tubulin intensity for CD70 in panel A seems weaker than for the other samples for some reason.

Reviewer #2:

1. The only very minor point I have is as follows: P10 li 202 Precision of KD values is overestimated, please round to one digit (e.g. 30.77 =30.8).

Reviewer #3:

2. Please explain the treatment scheme shown in Fig.1. Why were molecules used as single treatments and then in the middle as combination treatment. Why were not all treatments combination treatments?
3. When analysing T-regs in Fig. 1, it says tumors were harvested on day 20. Are these the same animals as shown in Fig.1c? How could some animals survive beyond day 20 when tumors were harvested?
4. Would it be possible to include epitope data in Suppl. Fig. 4?

Reviewer #1			
Major points			
Comment	Author response	Changes made	Line number/ Figure number
1. CD27:CD70 complex – crystal structure from recent publication	Thank you for raising this highly relevant point. The paper on the CD27:CD70 crystal structure was published after we started the submission and so was not taken into account. The manuscript has been updated to reflect this new data. Importantly, this new structure does not alter the projected binding sites of the antibodies.	We have exchanged the previous modelling of the CD27:CD70 complex based on the CD40:CD40L complex with the crystal structure published by Liu et al., 2021. Accordingly, the text and figure legend was changed to the following: Main text: The crystal structure of the CD27-CD70 trimeric complex has recently been determined (PDB:7KX0⁴⁷; Fig. 5g). The structure shows trimeric CD70 bound by three CD27 molecules, with CD70 interfacing CRD2 and CRD3 of CD27. This structure supports our site-directed mutagenesis analysis, with critical contacts for complex formation being identified in CRD2 of CD27. Based on CD70's epitope with CD27, internal binding was defined as CD70-facing, while external epitopes are on the opposite site of the receptor to the CD70 interface. Figure legend: Proposed model of the hCD27-CD70 trimer (PDB:7KX0) (top image: 'side' view, bottom image: 'top' view).	Lines 304-309 Lines 322-323

		 CD27-CD70 trimer	Fig. 5g
2. Consistency in amino acid numbering between Fig 4a and 4f/g	We apologise for the lack of clarity and have now adjusted the numbering of the schematic image of the CD27 receptor in Fig. 4a.	The numbering system of the cartoon has been adjusted to the numbers of the amino acids as shown in Supplemental Fig. 4a and the figure legend has been changed accordingly. Fig. 4a: CRD1: 39-75 CRD2: 76-116 CRD3: 117-153 Intracellular domain: 225-272  Figure legend: [...] and the cartoon shows the hCD27 homodimer and the number of amino acid residues in each CRD based on the sequence presented in Supplemental Fig. 4.	Fig. 4a Line 269-2709

3. The inability of hCD27.15 to block binding of CD70 and other mAb binding	Thank you for these insightful comments. We will address these comments in two parts. First, we agree that the rapid off-rate of hCD27.15 might account for why it does not block the binding of other mAb and CD70. We have amended the discussion to incorporate this. Second, the point raised about binding, affinity and saturation should similarly apply to the data obtained by Van Eenennaam et al. where the same hCD27.15 antibody and CD70 fusion protein was used. We hypothesise that discrepancies in the results might be accounted for by differences in experimental conditions, such as cell type used and related differences in CD27 receptor abundance on the cell surface. We have addressed and elaborated this in the main text.	The following has been added into the main text, addressing point one: One potential factor explaining the inability of hCD27.15 to block the binding of other mAb and CD70 is its high off-rate, which could allow other mAb to bind and out-compete hCD27.15. The following has been added in the main text, addressing the second point: The discrepancies in our observations might be due to differences in experimental conditions. Despite using the same mAb and recombinant mouse CD70 fusion protein concentrations, Van Eenennaam et al. employed CD27 transfected CHO-K1 cells which might express higher levels of CD27 than PBMC, potentially affecting the binding of hCD27.15.	Line 471-473 Line 474-478
4. Receptor dimerization and impact on binding epitope	This is a highly relevant point. We have altered the discussion to address this.	Our binding models display CD27 mAb binding to monomeric CD27. However, the binding epitopes may be altered by receptor dimerisation secondary to disulphide bonding or through the preligand binding assembly domain (PLAD) formation. To date, the exact arrangement of a CD27 homodimer is still unknown. We speculate that dimerisation of the receptor may reduce the accessibility of mAb binding sites, in particular those internal-facing and membrane-proximal epitopes.	Line 493-497
5. Additional information in antibody methods section	Thank you for raising this relevant point. The requested information has been added as follows in the Methods (Antibodies)	Anti-mCTLA4 m2b (9D9) was purchased from BioXCell. [...] Transient antibody production was performed using ExpiFectamine CHO Transfection Kit (gibco) and antibodies were purified using affinity chromatography. All antibodies were tested for endotoxin levels (<5	Line 586-589 (Methods)

on antibody production and quality control		EU/mg) and regularly checked for aggregation by HPLC and de-aggregated (if aggregation >1%) by size exclusion chromatography.	
6. Supplemental Fig. 3a – gating strategy for CD4 ⁺ and CD8 ⁺ T cells	Thank you for pointing this out. The flow cytometry staining panel did not include a CD8 marker and the population we assume to be CD8 ⁺ T cells might also contain other cell types such as NKT cells (CD3 ⁺ CD4 ⁺ CD56 ⁺). However, since this immune cell subset accounts for a small proportion of splenocytes (approximately 4%; Hammond et al., 2001, J Immunol; DOI: 10.4049/jimmunol.167.3.1164) we think that this does not have an impact on our interpretation.	We have addressed this comment and changed the figure legend of Supplemental Fig. 3a accordingly: Figure legend: (a) Gating strategy for CD4 ⁺ and presumed CD8 ⁺ T cells. (b) Shown are dot plots for CD4 ⁺ and presumed CD8 ⁺ T cells of spleens harvested on day 13 representative of two mice per treatment arm.	Supplemental Fig. 3a – figure legend
7. Supplemental Fig. 5b and c – correlations	This is a valid point. This has been re-worded in the results and discussion as advised.	Main text: [...] no significant correlations were observed. However, if hCD27.15 was removed as a visual outlier, a significant association was observed between affinity and activity, with the highest affinity mAb being the most agonistic (Supplementary Fig. 5). [...] but only after hCD27.15 was removed as an outlier from the correlation analysis.	Supplementary Fig. 5b and c Line 340-342 Line 481-482
Minor points			
1. Line 89 – typo	Thank you for bringing this to our attention.	“determinates” changed to “determinants”	Line 88

2. Line 119-120	Thank you for this helpful suggestion. We have changed this accordingly.	Main text: Altogether, these results support the general applicability of using agonistic anti-CD27 to enhance the effects of Treg-depleting mAb in different tumor models.	Line 118-119
3. Tumour harvest day 20 (Fig.1)	We apologise for the confusion and have altered the legend in Fig 1.	Figure legend: d-f, CT26-bearing mice were treated as described in (a). Tumors were harvested on day 20 and (d) %Tregs and (e) %CD8+ T cells and (f) CD8/Treg ratio was determined.	Line 128-1290 Fig. 1d-f
4. Line 164 – monocyte numbers at day 13	We apologise for this oversight and the text describing the change in monocytes numbers on day 13 have been altered.	Main text: [...] In contrast, treatment with m2a led to a significant reduction of monocytes on d13. Anti-mCD27 m1 also resulted [...]	Line 163-164
5. 197 – K _d as avidity	Thank you for pointing this out and we have changed the text as suggested.	Main text: To assess bivalent mAb binding avidity, surface plasmon resonance (SPR) was employed (Fig. 3c and 3d).	Line 202
6. Semi-log scale hCD27 mAb binding (Fig 3b)	Thank you for this very helpful suggestion. The x-axis of Fig. 3b has been changed to a log axis.		Fig. 3b
7. Line 239 – mentioning AT133-14	We agree, this would be useful information.	The main text has been changed accordingly. Main text: CD70's binding was poorly defined and either partially or fully blocked by a number of mAb (CRD1-binding AT133-5 and AT133-11, CRD2 and CRD3-binding varli and CRD3-binding AT133-14).	Line 240
8. Line 250 or 259? – listing	Whilst amino acid residues I110 and T111 appeared as crucial epitopes for	Alanine scanning data for the single mutation of residues I110 and T111 have been added in the heatmap (additional bottom two rows) in Fig	Fig. 4g

additional residues for hCD27.15	hCD27.15 in the pairwise mutation, analysis, the single mutation of these amino acids did not result in loss of mAb binding and therefore were not considered as crucial for binding. We have data supporting this and apologise to have not included it in the manuscript.	4g. The data show that both amino acids are not crucial for mAb binding.	
9. Line 274 – change of wording	Thank you for your suggestion. We have reworded accordingly.	Main text: The heatmap shows the ratio of MFI achieved after PBMC were incubated with the competing reagent, and after PBMC were treated with the respective isotype control.	Line 275-277
10. Fab and F(ab)	Thank you, now changed.	Fab changed to F(ab) throughout the manuscript	Line 59, 292, 298 Fig. 5a
11. Fv or Fab??	Thank you for bringing this to our attention. We agree and the changes have been made.	The sentence reads now as follows: Figure legend Fig. 5h: h, Composite figure of the CRD1-binding Fv, [...]	Line 325-326 Fig. 5h
12. 357 – GFP expression for hCD27.15 h2	Thank you for raising this. We agree, this hasn't been mentioned in the text and will be adjusted,	GFP expression of hCD27.15 h2 was included in the text as follows: Main text: Across all mAb, the h2 isotype induced the highest GFP expression for 4 out of 6 mAb (varli: 34.9%, hCD27.15: 59.7%, AT133-2: 41.8%, AT133-11: 31.9%). However, this was only statistically significant for hCD27.15.	Line 359

13. Fig. 6 – Changing figure title	We do agree with this comment and thank the reviewer for their helpful suggestions.	The title of Fig.6 has been changed to: Fig. 6 Influence of Fc format on agonism.	Line 399 Fig. 6
14. Line 410 - typo	Thank you for bringing this to our attention.	The repeated word “show” has been removed and the sentence reads now as follows: Graphs (b-d) show means + SD and (e-g) median with ranges.	Line 412
15. Line 428 – AT133-14 vs AT133-2	We apologise for this oversight and have corrected it.	Main text: A few small clusters were observed on the surface after treatment with varli h1, AT133-2 h1, AT133-14 h1 and CD70, but hCD27.15 h1 evoked larger clusters and the highest number of clusters on the cell surface (hCD27.15 vs varli or AT133-14: 3-fold increase, hCD27.15 vs AT133-2: 2-fold increase in numbers of cluster per cell) (Fig. 7d).	Line 431
16. Citation of Fig. 7d	Thank you for your suggestion.	Fig. 7d has been added: To examine if this was the case, we stimulated hCD27-GFP transfected Jurkat cells with hCD27 h1 mAb (Fig. 7a, 7c and 7d). [...] Next, we employed, h2 isoforms of the same mAb (Fig. 7b, 7c and 7d).	Lines 427 and 432
17. 512 – rewording of “poorly understood” to “lacking”	We thank the reviewers for the suggestion to reword this sentence.	“Poorly understood” has been replaced with “lacking” and the sentence reads as follows: Main text: Their clinical efficacy is modest and definitive evidence of agonism, or indeed, understanding behind their mechanism of action, is lacking.	Line 523
18. Methods for CT26 mice experiments in “Tumor models” section	We apologise for the omission of this information and have now included it.	Methods for CT26 tumor experiments have been included in the methods: Main text: For CT26 tumor experiments, BALB/c mice were subcutaneously injected with 5x10 ⁵ CT26 colon carcinoma cells at day 0 followed intraperitoneal administration of 200 µg anti-mCTLA-4 m2b (9D9) on day 10, 13, 16 and 19 or 100 µg anti-mCD27 m1 (AT124-1) on day 11, 13, 16 and 18 or the combination.	Line 564-5671

19. Supplementary figure references	Thank you for bringing this to our attention.	We have addressed this comment and removed specific cross-citations to specific sub-figures of figure panels in the supplementary figure legends.	-																								
20. Supplementary Fig. 7b – inclusion of CD70 in densitometry graph	Thank you for pointing this out. We agree, it would be useful to have the investigated CD27 antibodies in comparison with CD70.	The requested data about CD70 has been added to the graph. 	Supplementary Fig. 7b																								
Reviewer #2																											
1. K_D values	Thank you for raising this.	The K_D values in the main text and the according Fig. 3 were changed to one digit.    k_a ($\times 10^5$) 2.2 2.4 3.7   k_d ($\times 10^{-4}$) 6.7 74.2 21.0   K_D ($\times 10^{-9}$) 3.0 30.8 5.7   k_a ($\times 10^5$) 0.6 0.6 8.0   k_d ($\times 10^{-4}$) 16.5 32.3 5.0   K_D ($\times 10^{-9}$) 28.2 50.4 0.6   	k_a ($\times 10^5$)	2.2	2.4	3.7	k_d ($\times 10^{-4}$)	6.7	74.2	21.0	K_D ($\times 10^{-9}$)	3.0	30.8	5.7	k_a ($\times 10^5$)	0.6	0.6	8.0	k_d ($\times 10^{-4}$)	16.5	32.3	5.0	K_D ($\times 10^{-9}$)	28.2	50.4	0.6	Line 202-204 and Fig. 3
k_a ($\times 10^5$)	2.2	2.4	3.7																								
k_d ($\times 10^{-4}$)	6.7	74.2	21.0																								
K_D ($\times 10^{-9}$)	3.0	30.8	5.7																								
k_a ($\times 10^5$)	0.6	0.6	8.0																								
k_d ($\times 10^{-4}$)	16.5	32.3	5.0																								
K_D ($\times 10^{-9}$)	28.2	50.4	0.6																								
Reviewer #3																											
1. Explanation of treatment scheme Fig. 1	Thank you for requesting clarification on this. The chosen treatment schedules are based on our experience with the monotherapies in this tumor model. For each mAb, we have continued to adhere to the	No changes made.	Line 124-125 Fig. 1																								

	monotherapy scheduling, where the best tumor reduction is observed. The concurrent administration of anti-mCTLA-4 and anti-CD27 on days 13 and 16 are coincident.		
2. Tumour harvest day 20 (Fig.1)	Thank you for also raising this. We have responded to this above (Reviewer 1, Minor Point 3)		
3. Inclusion of epitope data in Supplementary Fig. 4	Apologies, but we do not fully understand this request. Can the reviewer please clarify which epitope data is being sought?		

REVIEWERS' COMMENTS:

Reviewer #1 (Remarks to the Author):

The authors have addressed all my previous concerns and suggestions in a very organized manner. Excellent article.

Reviewer #3 (Remarks to the Author):

Dear authors have addressed and answered the issues raised by the reviewers. Regarding question 3 of reviewer 3, binding regions are shown in Figure 4 and 5 and I thought it might be possible to transfer this information to the sequence file. However, I agree that it is presumably sufficient as shown in Figure 4 and 5. Thus, I support publication of the manuscript.